



# Benchmarking $K_{DP}$ in Rainfall: a Quantitative Assessment of Estimation Algorithms Using C-Band Weather Radar Observations

Miguel Aldana[1], Seppo Pulkkinen[1], Annakaisa von Lerber[1], Matthew R. Kumjian[2], and Dmitri Moisseev[1, 3]

[1]Space Research & Observation Technologies, Finnish Meteorological Institute, Helsinki, Finland
[2]Department of Meteorology & Atmospheric Science, The Pennsylvania State University, Pennsylvania, United States
[3]Institute for Atmospheric & Earth system Research, University of Helsinki, Helsinki, Finland

**Correspondence:** Miguel Aldana (miguel.aldana@fmi.fi)

**Abstract.** Accurate and precise $K_{DP}$ estimates are essential for radar-based applications, especially in quantitative precipitation estimation and radar data quality control routines. The accuracy of these estimates largely depends on the post-processing of the radar's measured $\Phi_{DP}$, which aims to reduce noise and backscattering effects while preserving fine-scale precipitation features. In this study, we evaluate the performance of several publicly available $K_{DP}$ estimation methods implemented in

open-source libraries such as PyArt and Wradlib, and the method used in the Vaisala weather radars. To benchmark these methods, we employ a polarimetric self-consistency approach that relates $K_{DP}$ to reflectivity and differential reflectivity in rain, providing a reference self-consistency $K_{DP}$ ($K_{DP}^{sc}$) for comparison. This approach allows for the construction of the reference $K_{DP}$ observations that can be used to assess the accuracy and robustness of the studied $K_{DP}$ estimation methods. We assess each method by quantifying uncertainties using C-band weather radar observations where the reflectivity values ranged

between 20 and 50 dBZ.

Using the proposed evaluation framework we could define optimized parameter settings for the methods that have user-configurable parameters. Most of such methods showed significant reduction in the estimation errors after the optimization with respect to the default settings. We have found significant differences in the performances of the studied methods, where the best performing methods showed smaller normalized biases in the high reflectivity values (i.e., $\geq 40$ dBZ) and overall

smaller normalized root mean squared errors across the range of reflectivity values.

## 1 Introduction

The specific differential phase ($K_{DP}$) plays an important role in many weather radar applications, particularly in hydrometeor classification (Vivekanandan et al., 1999; Liu and Chandrasekar, 2000; Zrnić et al., 2001; Keenan, 2003; Lim et al., 2005; Tessendorf et al., 2005; Marzano et al., 2007; Dolan and Rutledge, 2009; Park et al., 2009; Snyder et al., 2010; Al-Sakka et al.,

2013; Dolan et al., 2013; Thompson et al., 2014; Bechini and Chandrasekar, 2015; Grazioli et al., 2015; Wen et al., 2015; Besic et al., 2016; Ribaud et al., 2019) and quantitative precipitation estimation (QPE) (Sachidananda and Zrnić, 1987; Chandrasekar et al., 1990; Ryzhkov and Zrnić, 1995, 1996; May et al., 1999; Bringi and Chandrasekar, 2001; Bringi et al., 2006; Matrosov et al., 2006; Giangrande and Ryzhkov, 2008; Bringi et al., 2011; Cifelli et al., 2011; Wang et al., 2013; Chen and Chandrasekar,





2015; Chen et al., 2017; Thompson et al., 2018; Zhang et al., 2020), and is used in data assimilation for numerical weather

prediction models (Thomas et al., 2020; Du et al., 2021) and in hydrological applications (Brandes et al., 2002; Ryzhkov et al., 2005b; Vulpiani et al., 2012; Li et al., 2023; Cremonini et al., 2023). Compared to radar power variables, i.e., reflectivity factor at horizontal polarization ($Z_H$) and differential reflectivity ($Z_{dr}$), $K_{DP}$ offers advantages in terms of accuracy, resilience, and reliability due to its immunity to radar miscalibration, attenuation (Bringi and Chandrasekar, 2001; Illingworth, 2004; Ryzhkov and Zrnic, 2019), and partial beam blockage (Zrnić and Ryzhkov, 1996). It has also proven successful in hydrometeor classifi-

cation routines (Lim et al., 2005; Park et al., 2009; Grazioli et al., 2015; Tiira and Moisseev, 2020), especially in the detection of graupel (Dolan and Rutledge, 2009; Oue et al., 2015), small melting hail (Kumjian et al., 2019), and dendritic growth zone and processes within (Kennedy and Rutledge, 2011; Andrić et al., 2013; Schneebeli et al., 2013; Moisseev et al., 2015; Kumjian and Lombardo, 2017). The ability of $K_{DP}$ to accurately estimate heavy rainfall, differentiate hydrometeor types, and overcome attenuation in precipitation makes it an invaluable operational and research radar variable.


Despite its advantages, accurate estimation of $K_{DP}$ from radar-measured differential phase ($\Phi_{DP}$) remains challenging. Mathematically, $K_{DP}$ is half the range derivative of $\Phi_{DP}$, which measures the phase shift between horizontally and vertically polarized signals as they propagate through precipitation. This phase shift ($\Phi_{DP}$) is influenced by hydrometeor concentration, shape, orientation, and composition (Kumjian, 2018). However, $\Phi_{DP}$ is not typically smooth and monotonically increasing

along the rain path; it contains fluctuations due to noise ($\epsilon$) and backscattering differential phase ($\delta_{HV}$) (Ryzhkov and Zrnić, 1996; Ryzhkov and Zrnic, 1998). Excessive filtering of $\Phi_{DP}$ to remove $\epsilon$ can lead to the loss of fine-scale precipitation features, affecting the accuracy of $K_{DP}$ estimates especially in light precipitation (Huang et al., 2017). In heavier precipitation, $\delta_{HV}$ causes spikes in $\Phi_{DP}$, especially at higher radar frequencies, further complicating accurate $K_{DP}$ estimation (Bringi and Chandrasekar, 2001).


To address these challenges, various methods have been developed to post-process $\Phi_{DP}$ and derive $K_{DP}$ (Hubbert et al., 1993; Hubbert and Bringi, 1995; Ryzhkov et al., 2005c; Wang and Chandrasekar, 2009; Otto and Russchenberg, 2011; Maesaka et al., 2012; Vulpiani et al., 2012; Schneebeli and Berne, 2012; Giangrande et al., 2013; Schneebeli et al., 2014; Huang et al., 2017; Reinoso-Rondinel et al., 2018; Wen et al., 2019). Basic approaches include median filters and moving windows,

while more advanced methods use regression techniques and self-consistency constraints based on $Z_H$ or $Z_{dr}$. Many of these methods are now available in open-source Python libraries such as PyArt (Helmus and Collis, 2016) and Wradlib (Heistermann et al., 2013). For this study, some of the most popular implemented methods based on Maesaka et al. (2012), Vulpiani et al. (2012), Giangrande et al. (2013), and Schneebeli et al. (2014) were selected for analysis. Additionally, the $K_{DP}$ product implemented by Vaisala in IRIS software (Vaisala, 2017) based on Wang and Chandrasekar (2009) was also included in our

analysis. Each algorithm has its own data requirements, mathematical approach, and optimizing parameters, raising the question of which performs optimally under varying parameter settings, and rainfall intensities.

Recent studies show that $K_{DP}$ estimates can vary significantly depending on the algorithm and the optimizing parameters

used. Reimel and Kumjian (2021) evaluated the errors of several methods using synthetic $K_{DP}$ profiles and found that no
single algorithm was optimal across all rainfall conditions. Instead, performance varied with the complexity of the rain profile
and selected parameters. They identified kdp_maesaka (PyArt's implementation of Maesaka et al. (2012)) and phase_proc_lp
(PyArt's implementation of Giangrande et al. (2013)) as particularly versatile. However, Reimel and Kumjian (2021) used
synthetic data, which may miss some of the effects present in radar observations of rainfall (e.g., $\delta_{HV}$). More recently, Li et al.
(2023) compared kdp_maesaka and phase_proc_lp in an extreme summer rainfall event, finding that fine-tuning the methods
played a key role in retrieving the most accurate $K_{DP}$ estimate. Despite these insights, the performance and uncertainties of
most methods in rainfall observations remain largely unexplored.

The goal of this study is to evaluate the performance of publicly available $K_{DP}$ estimation methods in real rainfall obser-
vations and quantify their uncertainties as a function of reflectivity intensities. We use self-consistency relations linking $K_{DP}$
to $Z_H$ and $Z_{dr}$ to compute benchmark $K_{DP}$ profiles (herein $K_{DP}^{sc}$), requiring thorough selection and filtering of data. $K_{DP}^{sc}$
computed from quality-controlled $Z_H$ and $Z_{dr}$ measurements provides a solid benchmark against which to compare the meth-
ods' performances, select optimal parameters, and quantify the associated uncertainties.

This paper is organized as follows: Section 2 describes the radar and disdrometer data, the evaluation framework, and intro-
duces the $K_{DP}$ estimation methods. Section 3 presents and discusses the parameter optimization and performance evaluation
of the methods, and Section 4 summarizes the findings.

## 2 Data & Methods

### 2.1 Radar & Disdrometer Data

This study evaluates the performance of $K_{DP}$ estimation methods using real rainfall data. The dataset was collected from the
Finnish Meteorological Institute's (FMI) C-band Vantaa radar, located near Helsinki, Finland (see Fig. 1). The radar recorded
various quantities, including $Z_H$, $Z_{dr}$, $\Phi_{DP}$, $K_{DP}$, cross-correlation coefficient ($\rho_{HV}$), and the hydrometeor classification
product available in IRIS (Vaisala, 2017) and based on the methodology described by Chandrasekar et al. (2013). The spatial
resolution of the radar is 500 meters in range and $1°$ in azimuth, with scans performed every 5 minutes, and the data was
collected with an elevation angle of $0.7°$. The dataset spans from June to September during the years 2017 to 2019, capturing
precipitation events with variable rainfall intensities and spatial extents. The raw radar dataset as well as the post-processed
$K_{DP}$ estimates are available from the link provided in Aldana (2024).

To ensure data quality, only periods when the Vantaa radar had calibration errors within 1 dB were selected. The calibra-
tion was verified by i) identifying periods where solar flux estimates from Vantaa radar estimates aligned consistently with
Dominion Radio Astrophysical Observatory (DRAO) estimates (Huuskonen and Holleman, 2007; Tapping, 2013; Holleman
et al., 2022), and ii) selecting radar scans within these periods where $Z_H$-calibration offsets were within 1 dB, following the





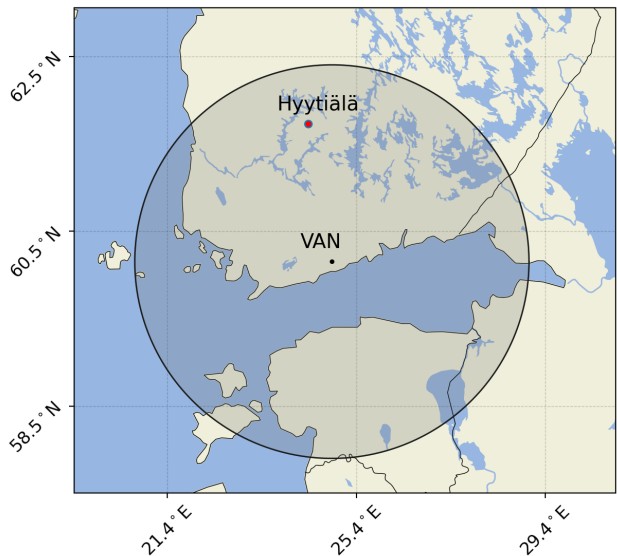

**Figure 1.** Map showing locations of FMI's Vantaa radar (VAN) and Hyytiälä's research station where DSD data were collected. The shaded area is a circle of 250 km radius corresponding to the spatial coverage of the radar.

absolute calibration procedure outlined by (Gourley et al., 2009). $Z_{dr}$ bias was estimated and corrected during these periods by computing the offset between observed and self-consistency $Z_{dr}$, derived from observed $Z_H$, as described in (Hickman, 2015) and computing the average for several cases.


The performance of the $K_{DP}$ estimation methods is benchmarked against self-consistency $K_{DP}^{sc}$, computed from measured $Z_H$, $Z_{dr}$ and using self-consistency relations. These relations, which link the polarimetric radar variables, have been widely used for radar calibration correction (Gorgucci et al., 1992; Goddard et al., 1994; Illingworth and Blackman, 2002; Vivekanan-dan et al., 2003; Ryzhkov et al., 2005a). The self-consistency relations were derived by fitting radar variables computed using the open-source library, PyTMatrix (Leinonen, 2014). PyTMatrix provides a simple interface for T-Matrix electromagnetic scattering calculations (Waterman, 1965; I. Mishchenko et al., 2000), requiring the user to provide drop size distribution (DSD) data and setting parameters such as temperature, radar wavelength's band and raindrop shape model. The settings applied were 10°C, C-band and Thurai et al. (2007), respectively, and the DSD data provided was collected by an optical Parsivel disdrom-eter (Moisseev, 2024) located in Hyytiälä, Finland (see Fig. 1).


The Parsivel disdrometer records the number of particles and velocity at one-minute intervals, sorting the data into 32 bins depending on particle's size (i.e., equivalent volume diameter) and 32 additional bins depending on particle's fall velocity. From the number of particles, size and velocity classes, the Parsivel disdrometer computes the precipitation type, which was used to filter out non-liquid particles. Observations were further limited to times when the 30-minute average 2-meter tem-





perature exceeded 2°C to ensure liquid rain only. Following the filtering procedure to reduce statistical errors suggested by Leinonen et al. (2012), only those measurements with at least 100 counts in two consecutive bins and positive counts in at least four consecutive bins were retained. The disdrometer dataset, covering June to September from 2014 to 2019, provided a robust basis for deriving average summer-season DSD parameters such as mean volume diameter ($D_0$), intercept parameter ($N_w$), and shape parameter ($\mu$). These parameters showed strong agreement with those reported by Leinonen et al. (2012) in

a climatological study of Finland. From the derived DSD parameters ($N_w$, $D_0$, and $\mu$), the polarimetric radar variables were computed and used for deriving the self-consistency relation defining the framework to evaluate the $K_{DP}$ estimation methods.

## 2.2   $K_{DP}$ Evaluation Framework

The performance of $K_{DP}$ estimation methods is evaluated using $K_{DP}^{sc}$ as benchmark. This quantity is calculated from each radar-measured tuple ($Z_H$, $Z_{dr}$) following a relationship of the form (Gourley et al., 2009):

$$K_{DP}^{sc} = z_H \times 10^{-5} \times (a_1 + a_2 \times Z_{dr} + a_3 \times Z_{dr}^2 + a_4 \times Z_{dr}^3), \tag{1}$$

where $z_H = 10^{0.1 \times Z_H}$ represents $Z_H$ in linear units (mm$^6$m$^{-3}$) and $Z_{dr}$ is in decibels (dB). The coefficients used in this relation are $a_1 = 6.78$, $a_2 = -2.65$, $a_3 = 0.562$ and $a_4 = -0.0624$. The coefficients align well with those reported by Gourley et al. (2009), which employed the raindrop shape models by Brandes et al. (2002) and Thurai and Bringi (2005).

To ensure the accuracy and robustness of $K_{DP}^{sc}$ estimates used in the method assessment, it was crucial to quality-control the $Z_H$ and $Z_{dr}$ data. Radar observations of rain are often affected by non-meteorological measurements, resonance effects, and hail contamination (Bringi and Chandrasekar, 2001; Kumjian, 2013; Ryzhkov and Zrnic, 2019). To address these issues, the following filtering steps were applied:

  – *Noise filtering*: A minimum threshold of 0.97 was applied to $\rho_{HV}$.

– *Non-meteorological observations filtering*: The hydrometeor classification product from IRIS (Vaisala, 2017), based on Chandrasekar et al. (2013), was used to exclude gates classified as non-meteorological.

  – *$\delta_{HV}$ reduction*: Gates with $Z_{dr} > 3.5$ dB were excluded (Bringi and Chandrasekar, 2001; Gourley et al., 2009).

  – *Non-liquid rain filtering*:

    – Only radar scans from the warm months (Jun-Sep) were selected.

– Gates not classified as rain by the hydrometeor classification product were excluded.

    – Hail contamination was addressed by removing gates with $Z_H \geq 50$ dBZ.

    – Observations from the melting layer and above were suppressed by masking gates farther than 70 km (see last dashed ring in Fig. 2) from the radar in the radial direction. The distance was manually set by identifying gates with melting layer signatures (Giangrande et al., 2008; Boodoo et al., 2010).



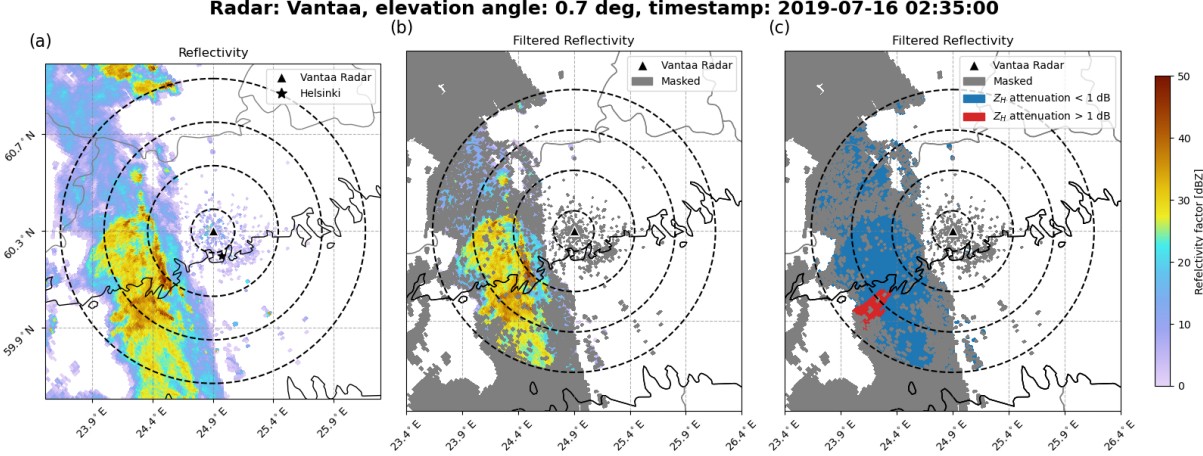

**Figure 2.** Example of a Vantaa radar scan during a precipitation event on the 16 July 2019 at an elevation angle of $0.7°$. Panel (a) shows measured $Z_H$; panel (b) shows filtered $Z_H$ with masked gates in grey; panel (c) shows the same as (b) but with attenuated gates marked in red and non-attenuated gates marked in blue. Dashed rings represent radial distances of 10, 30, 50 and 70 km from the radar.

In addition to addressing noise and non-liquid rain measurements, $K_{DP}^{sc}$ estimates are affected by attenuation in $Z_H$ and differential attenuation in $Z_{dr}$, particularly in cases of heavy rainfall, extended propagation paths through rain (hereafter "rain paths") (Zrnić and Ryzhkov, 1996; Carey et al., 2000; Bringi and Chandrasekar, 2001; Kumjian, 2013), and when the radar's antenna radome is wet (Blevis, 1965; Kurri and Huuskonen, 2008). To mitigate these effects, radar scans when there was rain on top of the radar within the past 20 minutes were discarded. Then, for the remaining cases, attenuation in heavy precipitation

or extended rain paths were addressed by flagging the radar gates when suspected attenuation of at least 1 dB was detected. The attenuation in range gates was inferred using a standard method that linearly relates the losses in $Z_H$ and $Z_{dr}$ with increases in $\Delta\Phi_{DP}$ (Ryzhkov and Zrnić, 1995; Carey et al., 2000; Bringi and Chandrasekar, 2001; Gourley et al., 2009). $\Delta\Phi_{DP}$ corresponds to the total span of $\Phi_{DP}$ along the radial within a rain path. A rain path was defined as a set of consecutive gates with rain features extending at least 20 km in the radial direction. For C-band radar, a minimum threshold of $12°$ in $\Delta\Phi_{DP}$

indicates attenuation of at least 1 dB (Carey et al., 2000). In this study, a threshold of $10°$ was used, meaning that gates within rain paths featuring $\Delta\Phi_{DP} \geq 10°$ were flagged as attenuated.

An example of the filtering procedure applied to a radar scan is shown in Fig. 2. This figure demonstrates the effects of the filtering process and considered attenuation on the chosen data samples.


    Following the filtering process, the dataset comprised 652,624 quality-controlled gates from 70 radar scans. Figure 3 presents a histogram of the data proportions across different $Z_H$ values, showing the highest percentage of data between 30-35 dBZ, with a sharp decrease from 35-50 dBZ. The stacked bars indicate the percentages of attenuated and non-attenuated gates, with





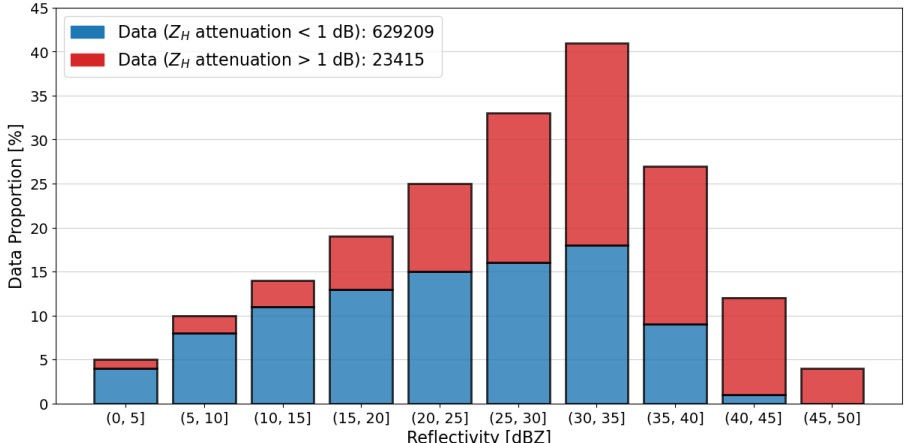

**Figure 3.** Proportion of data across $Z_H$ intervals of 5 dBZ. Attenuated data is represented by red bars and non-attenuated data is represented by blue bars. The legend indicates the total number of gates with suspected attenuation of at least 1 dB (red) and less than 1 dB (blue).

the ratio of attenuated to non-attenuated data increasing with greater $Z_H$.


## 2.3 $K_{DP}$ Estimation methods

This Section provides an overview of the $K_{DP}$ estimation methods selected for this study. The selection criteria focused on the availability of these methods in widely used open-source libraries, such as PyArt (Helmus and Collis, 2016) and Wradlib (Heistermann et al., 2013). At the time of this study, PyArt version 1.17.0 included the following methods: kdp_maesaka,

kdp_vulpiani, phase_proc_lp and kdp_schneebeli. Wradlib version 2.0.3 included kdp_from_phidp and phidp_kdp_vulpiani. However, phidp_kdp_vulpiani was excluded from our analysis, as it is based on the same method proposed by Vulpiani et al. (2012) that is already represented by PyArt in kdp_vulpiani. Additionally, kdp_iris, a method based on Wang and Chandrasekar (2009) and implemented by Vaisala in the IRIS software (Vaisala, 2017) was included. Table 1 summarizes the key features of the selected methods and a brief description of the methods is provided below.

a. **kdp_maesaka**. Developed by Maesaka et al. (2012) and available in PyArt, this method estimates non-negative $K_{DP}$ from liquid precipitation measurements. It addresses the issue of negative $K_{DP}$ estimates observed in exclusively liquid precipitation regions when using classical methods based on iterative filtering and local linear regression. Maesaka et al. (2012) identified that negative $K_{DP}$ were caused by noise in $\Phi_{DP}$ during weak precipitation and $\delta_{HV}$ during heavy precipitation. The method restricts $K_{DP}$ to positive values and assumes that $\Phi_{DP}$ is a monotonically increasing function 175     with range, already unfolded.

    b. **kdp_vulpiani**. Developed by Vulpiani et al. (2012) and available in PyArt, this method estimates $K_{DP}$ for any type of precipitation. It uses a multistep, moving-window range derivative approach to obtain $K_{DP}$. It calculates a $K_{DP}$





**Table 1.** List of $K_{DP}$ methods studied with key features.

| Method | Source | Data pre-requisites | Precipitation Type | Mathematical Approach (constraints) | Tested Parameters |
|---|---|---|---|---|---|
| kdp_maesaka | PyArt | Unfolded $\phi_{DP}$ | liquid | Variational | *clpf* |
| kdp_vulpiani | PyArt | Pre-filtered $\Psi_{DP}$ | Any | Moving window | *windsize*, *n_iter* |
| phase_proc_lp | PyArt | Unattenuated $Z_H$ | liquid | Linear Programming ($K_{DP}(Z_H)$) | *self_const*, *coef*, *window_len* |
| kdp_from_phidp | Wradlib | No NaN values | Any | Moving window | *winlen*, *dr* |
| kdp_schneebeli | PyArt | Pre-filtered $\Psi_{DP}$ | Any | Kalman filter | - |
| kdp_iris | IRIS | - | Any | Adaptive regression | - |

profile from the range derivative of a noise-reduced, offset-corrected, and unfolded $\Phi_{DP}$ profile. At each window, $K_{DP}$ is compared to thresholds representing unrealistic $K_{DP}$ values within precipitation, correcting possible aliasing with the

minimum threshold.

c. **phase_proc_lp**. Developed by Giangrande et al. (2013) and available in PyArt, this method estimates non-negative $K_{DP}$ from liquid precipitation measurements. It uses a linear-programming (LP) method to enforce monotonic behavior in $\Phi_{DP}$, restricting $K_{DP}$ to positive values. It extracts $\delta_{HV}$ from $\Phi_{DP}$ and it uses self-consistency constraints to bound $K_{DP}$ estimates based on measured $Z_H$. The method requires quality-controlled $Z_H$ and allows user-defined thresholds

to exclude hail and set the environmental 0 °C level to exclude mixed-phase particles.

d. **kdp_from_phidp**. Implemented in wradlib (Heistermann et al., 2013) based on Vulpiani et al. (2012), this method estimates $K_{DP}$ for any type of precipitation. It computes range-wise differentiation of $\Phi_{DP}$ over a user-defined window size length, defaulting to 7 gates for a range resolution of 1 km. Unlike kdp_vulpiani, it allows the selection of the method for range-gate differentiation, albeit it does not support multiple iterations, prioritizing speed over phase unfolding and

noise issues in $\Phi_{DP}$.

e. **kdp_schneebeli**. Developed by Schneebeli et al. (2014) and available in PyART, this method estimates $K_{DP}$ for any type of precipitation. It selects the best-averaged $K_{DP}$ profile from forward and backward propagation Kalman-filtered estimates. The Kalman filters are applied twice to each range gate state (accounting for forward and backward propagation) multiple times, recalculating the covariance matrices each time to yield unique states, and the best estimate is

selected.

f. **kdp_iris**. Implemented in the Vaisala's software IRIS (Vaisala, 2017) and based on Wang and Chandrasekar (2009), this method estimates $K_{DP}$ for any type of precipitation. It computes $K_{DP}$ adaptively through piece-wise regression and a regularization framework that minimizes both smoothness in $\Phi_{DP}$ and regression errors. The regularization adapts based





on range variations in $K_{DP}$ and $\rho_{HV}$ measurements, preserving steep $\Phi_{DP}$ changes in high-intensity precipitation while
reducing variations in low-intensity precipitation.

## 3 Results

### 3.1 Parameter Optimization of methods

In this Section, we optimize the methods kdp_maesaka, kdp_vulpiani, phase_proc_lp and kdp_from_phidp, by quantifying the errors under varying parameter settings, and selecting the optimal values. First, a qualitative analysis is provided with the use
of $K_{DP}$ vs. $Z_H$ scatter plots, illustrating the relationship between estimated $K_{DP}$ (y-axis) and $Z_H$ (x-axis), and benchmarking against $K_{DP}^{sc}$ (dashed black line). Then, the errors of each method as a function of parameter setting and $Z_H$ is provided. To achieve this, the dataset was divided into six 5-dB intervals ranging from 20 to 50 dBZ; we then computed for each interval the root mean square error (RMSE) and mean error (herein bias) and normalized by the mean $K_{DP}^{sc}$ from each interval. The optimal parameters were selected based on the smallest averaged normalized RMSE (herein NRMSE) in the last three $Z_H$
intervals (i.e., 35-50 dBZ), prioritizing the accuracy of $K_{DP}$ estimates in high-intensity precipitation.

### 3.1.1 PyArt's Maesaka method

PyArt's implementation of Maesaka et al. (2012), kdp_maesaka, features the optimizing parameter $Clpf$, which regulates the low-pass filter in $\Phi_{DP}$. The low-pass filter controls the degree of smoothing of $\Phi_{DP}$, with higher $Clpf$ values producing smoother $\Phi_{DP}$ profiles. In kdp_maesaka, the default value of $Clpf$ is 1.0, and this value is scaled by the range resolution
of the radar to match the resolution of the constraints applied to $\Phi_{DP}$. The scaling is proportional to the fourth power of the range resolution of the radar, and if we were to compare to the values used in Maesaka et al. (2012), a value of 1.0 corresponds to $10^{10}$ for Vantaa radar's range resolution of 500 m. In Maesaka et al. (2012), $Clpf$ values from $10^9$ to $10^{13}$ were tested on one rainfall case using a 250-m range resolution X-band radar. Their results show that values closer to $10^{13}$ suppressed fine-scale precipitation features while producing a smooth and clean $K_{DP}$, whereas values closer to $10^9$ preserved fine-scale
features while substantially including more noise. These results lead us to test values from $10^8$ to $10^{15}$, corresponding to $10^{-2}$ and $10^5$ in kdp_maesaka accounting for Vantaa's radar range resolution. Figure 4 shows scatter plots of $K_{DP}$ estimates using kdp_maesaka as a function of $Z_H$ for different $Clpf$ values. All scatter plots show overall accurate and precise $K_{DP}$ estimates within the $Z_H$ range 0-30 dBZ. This result implies that the subset of $Clpf$ values studied produce sufficiently smoothed $\Phi_{DP}$ to reduce the impact of noise in light precipitation. However, the effects of excessive smoothing are observed at the range of
40-50 dBZ, where $K_{DP}$ noticeably underestimates $K_{DP}^{sc}$. By comparing the scatter plots from $Clpf = 10^{-2}$ to $Clpf = 10^5$ in the $Z_H$ interval 40-50 dBZ, the underestimation of $K_{DP}$ is stronger with increasing $Clpf$.

To capture the influence of $Clpf$ on the errors when estimating $K_{DP}$ as a function of precipitation intensity, Figs. 5(a) and 5(b) show NRMSE and normalized bias of $K_{DP}$ estimates with varying $Clpf$. The smaller and consistent NRMSEs in regions





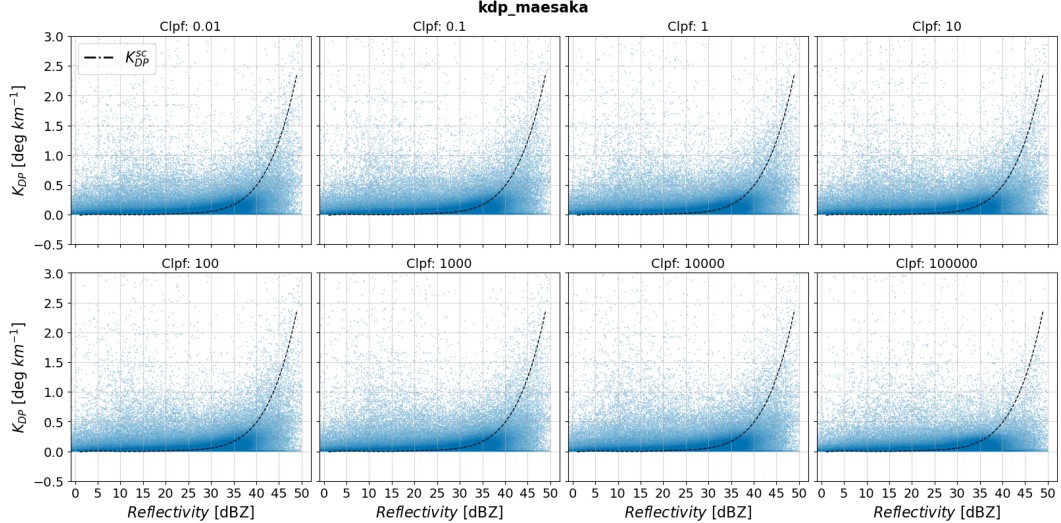

**Figure 4.** Scatter plots of estimated $K_{DP}$ from kdp_maesaka as function of reflectivity and for various values of $Clpf$. Panels (a)-(h) show results with $C_{lfp}$ values from $10^{-2}$ to $10^5$. The dashed black line corresponds to $K_{DP}^{sc}$.

of $Z_H \geq 35$ dBZ in Fig. 5(a) indicate that kdp_maesaka reaches stable solutions for all $Clpf$ values tested. However, $Clpf$ of $10^5$ showed the largest variability when transitioning from lowest to highest $Z_H$ among the values tested, producing largest NRMSE for $Z_H \geq 35$ dBZ and lowest otherwise. The underestimation of $K_{DP}$ using $10^5$ is evidenced in Fig. 5(b) for $Z_H \geq 35$ dBZ, where the results were the most negatively biased, whereas the results when from the remaining parameters produced consistent NRMSE values that were much smaller.


Our results show that larger values of $Clpf$ lead to larger errors due to oversmoothing in $\Phi_{DP}$. Overall, kdp_maesaka performs consistently when precipitation intensities reach 35 dBZ. The $Clpf$ yielding the smallest 35-50-dBZ-averaged NRMSE was $10^{-2}$.

### 3.1.2 PyArt's Vulpiani method

PyArt's implementation of Vulpiani et al. (2012), kdp_vulpiani, features two optimizing parameters: *windsize* (number of gates used for estimating $K_{DP}$) and *n_iter* (number of re-estimations of $K_{DP}$ per window). Higher values of these parameters result in smoother $\Phi_{DP}$ profiles. Reimel and Kumjian (2021) found various parameter combinations worked well depending on precipitation complexity, leading us to test combinations from 2 to 14 for both parameters. Figure 6 shows multiple scatter plots comparing the performance of kdp_vulpiani for different values of *windsize* and *n_iter* in estimating $K_{DP}$. The upper-

left corner panel shows the scatter of $K_{DP}$ using the largest tested settings, whereas the lower-right corner panel shows the results for the smallest. Each row holds *windsize* constant while each column holds *n_iter* constant. In the scatter plot for *windsize* = 14 and *n_iter* = 14, the data is predominantly clustered under $K_{DP}^{sc}$ for $Z_H \geq 35$ dBZ, indicating underestima-





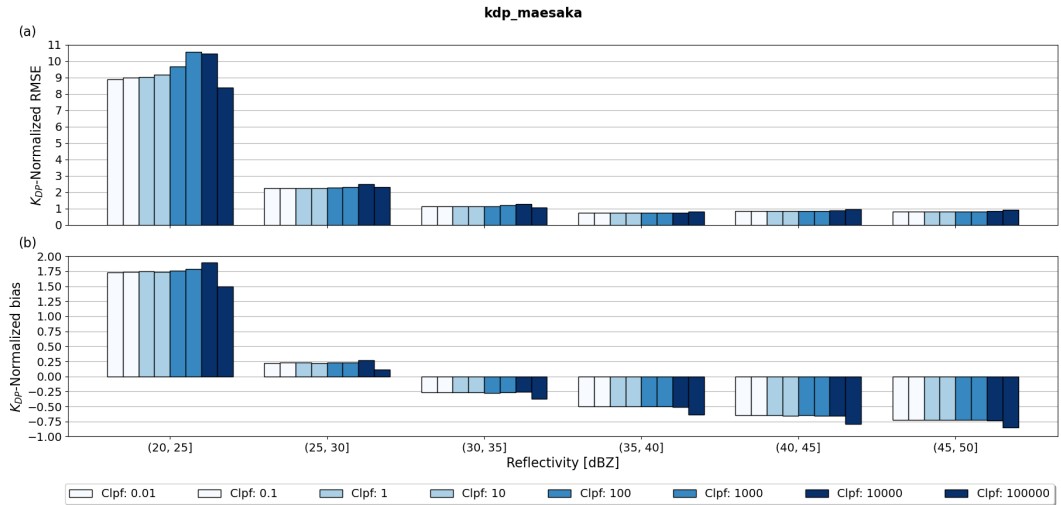

**Figure 5.** Panel (a) shows RMSE normalized by interval-averaged $K_{DP}^{sc}$ of kdp_maesaka relative to $K_{DP}^{sc}$ as a function of reflectivity and for various values of $Clpf$; panel (b) shows same as (a) but for the normalized bias metric.

tion of $K_{DP}$. For $Z_H < 35$ dBZ, this parameter setting produces accurate and precise results. The results from the opposite extreme of Fig. 6 are slightly more accurate albeit significantly less precise; the scatter plot of $windsize = 2$ and $n\_iter = 2$

shows wider spread of $K_{DP}$ data for all $Z_H$ values, although with slightly enhanced clustering of data around $K_{DP}^{sc}$ for $Z_H \geq 35$ dBZ. These results indicate a trade-off between precision and accuracy when varying $windsize$ and $n\_iter$ from 14 to 2. Particularly, larger settings favoring precision while deteriorating accuracy, and smaller settings favoring accuracy with deteriorated precision.

To further analyse the trade-off between accuracy and precision when varying $windsize$ and $n\_iter$ in kdp_vulpiani, Fig. 7(a)-(b) show the NRMSE and normalized bias of $k_{DP}$ estimates with varying $windsize$ and $n\_iter$ as a function of $Z_H$. Figure 7(a) shows that $windsize$ of 2 yielded the worst performance, implicating that the gain in accuracy by including fine-scale fluctuations in $\Phi_{DP}$ is not enough to compensate the increased errors due to the inclusion of outliers. On the other hand, $windsize$ of 14 shows good performance across the entire $Z_H$ range. However, the predominantly negative normalized bias of

$windsize$ of 14 relative to smaller counterparts in Fig. 7(b) indicates that larger $windsize$ leads to underestimation of $K_{DP}$ more than lower $windsize$ values. The consistent errors when varying $n\_iter$ in Fig. 7(a) indicate that this parameter setting does not impact as strongly as $windsize$ in the performance of kdp_vulpiani, especially in low $Z_H$. However, results from Fig. 7(b) suggest that smaller $n\_iter$ significantly reduces the underestimation of $K_{DP}$ estimates when $windsize$ is large. Our results strongly resemble those reported in Reimel and Kumjian (2021), indicating that smaller number of iterations and

moderate window sizes significantly enhance the performance of kdp_vulpiani. Particularly, among the RMSE heat maps of kdp_vulpiani shown in Reimel and Kumjian (2021), $windsize = 10$ and $n\_iter = 2$ produced the best results, coinciding with the smallest 35-50-dBZ-averaged NRMSE in this study.

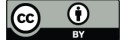

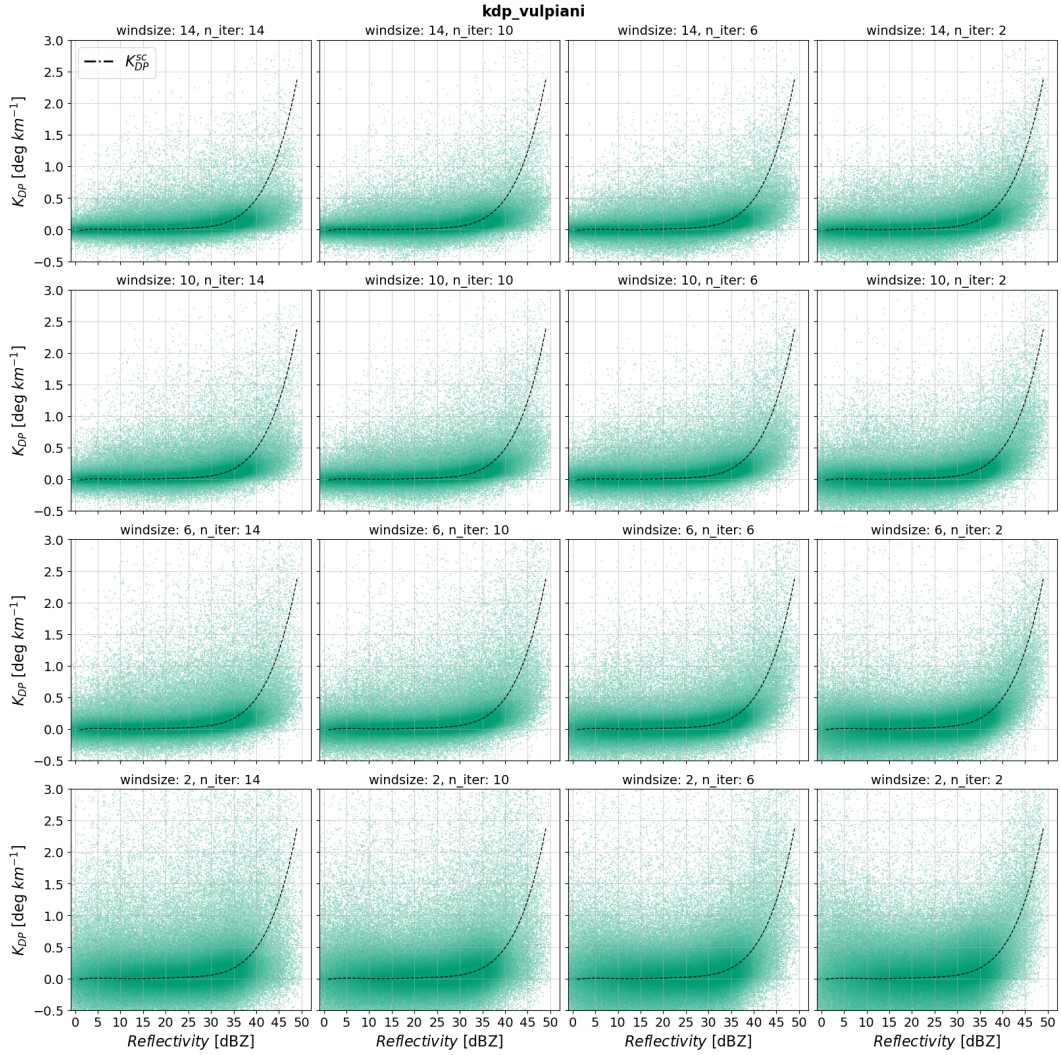

**Figure 6.** Scatter plots of estimated $K_{DP}$ from kdp_vulpiani as function of reflectivity and for various values of $windsize$ and $n\_iter$. Panels (a)-(p) show results with ($windsize$, $n\_iter$) tuples values from (14, 14) to (2, 2), decreasing $windsize$ with increasing rows. The dashed black line corresponds to $K_{DP}^{sc}$.

### 3.1.3 PyArt's Linear Programming method

PyArt's implementation of an LP method proposed in Giangrande et al. (2013), phase_proc_lp, allows the user to tune the window length for smoothing $\Phi_{DP}$, *window_len*, and two intertwined parameters constraining the $K_{DP}$ output via self-consistency relations: *self_const* and *coef*. The former is the weight of the self-consistency constraint and the latter is the exponent in the self-consistency relation linking $K_{DP}$ to $Z_H$, given in Giangrande et al. (2013) as $aZ_H^b$ but expressed in phase_proc_lp as $(10^{0.1 \times Z_H})^{coef}/self\_const$. Since information about the expected $K_{DP}$ was known beforehand, given by $K_{DP}^{sc}$, we pro-



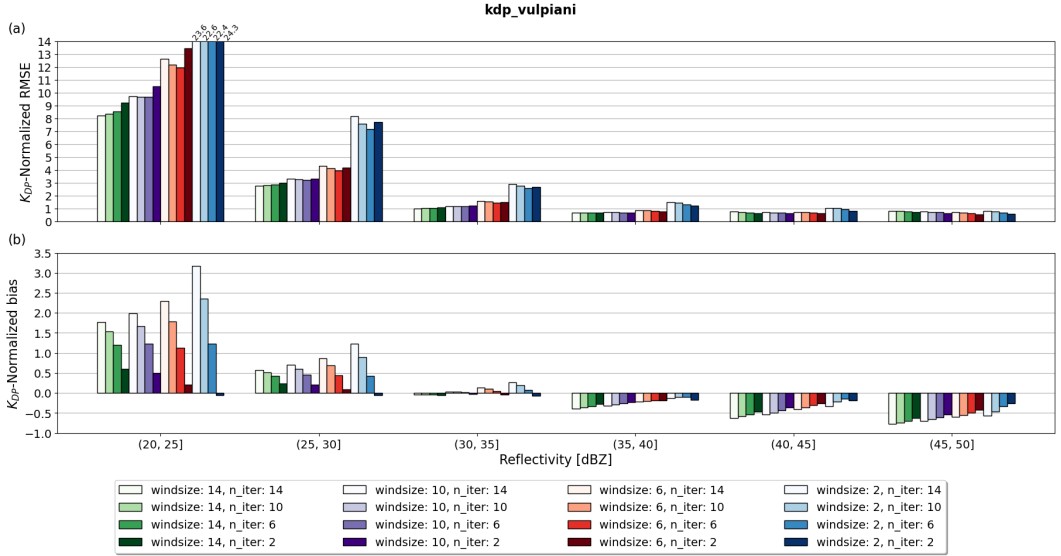

**Figure 7.** Panel (a) shows RMSE normalized by interval-averaged $K_{DP}^{sc}$ of kdp_vulpiani relative to $K_{DP}^{sc}$ as a function of reflectivity and for various values of $windsize$ and $n\_iter$; panel (b) shows same as (a) but for the normalized bias metric.

vided the method with the optimal values of $self\_const = 10^4$ and $coef = 0.914$. In this way, the parameter optimization of

phase_proc_lp was focused solely on $window\_len$ variations.

The parameter $window\_len$ defines the window length for smoothing the LP-processed $\Phi_{DP}$ field before $K_{DP}$ is estimated. The default setting of this parameter is 35, indicating a smoothing window length of 17.5 km for a range resolution of 500 m. To include finer-scale precipitation features (e.g. ∼2.5 km), phase_proc_lp was tested with $window\_len$ values ranging from 5

to 40. Figure 8 shows multiple scatter plots comparing the performance of phase_proc_lp for different settings of $window\_len$ in estimating $K_{DP}$. Moving from the upper-left to the lower-right corner panels, Fig. 8 shows $K_{DP}$ estimated using window lengths from 5 to 40 in intervals of 5. The scatter plot for $window\_len = 5$ shows data points predominantly clustered around $K_{DP}^{sc}$ across the entire $Z_H$ range, indicating strong correlation between $K_{DP}$ and $K_{DP}^{sc}$. Even in high $Z_H$ ranges (i.e., $\geq 35$ dBZ), the tight correlation between $K_{DP}$ and $K_{DP}^{sc}$ holds, indicating high accuracy and precision of $K_{DP}$ in the presence of

heavy precipitation. The accuracy and precision of $K_{DP}$ relative to $K_{DP}^{sc}$ decreases progressively when $window\_len$ increases, indicated by the spreading and downward shifting of $K_{DP}$ estimates relative to $K_{DP}^{sc}$. Especially for the range $Z_H \geq 35$ dBZ, the scatter plots of $windows\_len$ from 25 to 40 show substantial underestimation of $K_{DP}$ relative to $K_{DP}^{sc}$, indicating stronger oversmoothing of $\Phi_{DP}$ for larger values of $window\_len$. Comparing the scatter plots, $window\_len = 5$ undoubtedly shows the best performance of phase_proc_lp. This result agrees with phase_proc_lp $window\_len$ experiments by Li et al. (2023) in

an extreme heavy precipitation event, where small $window\_len$ yielded the best performance. Compared to the phase_proc_lp experiments by Reimel and Kumjian (2021), our results suggest that smaller $window\_len$ produce overall more accurate $K_{DP}$



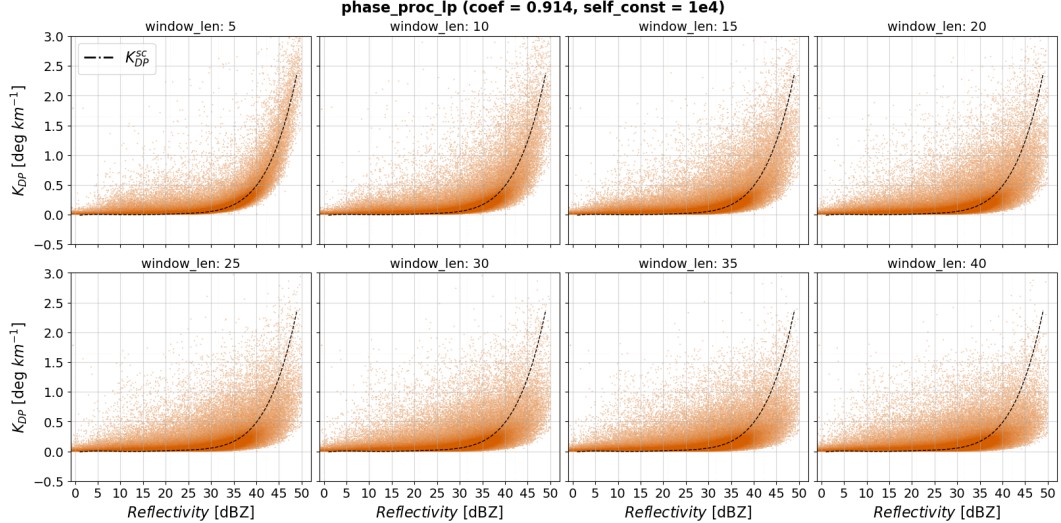

**Figure 8.** Scatter plots of estimated $K_{DP}$ from phase_proc_lp as function of reflectivity and for various values of $window\_len$. Panels (a)-(h) show results with $window\_len$ values from 5 to 40, while fixing $coef$ to 0.914 and $self\_const$ to $10^4$. The dashed black line corresponds to $K_{DP}^{sc}$.

estimates. However, the influence of the self-consistency constraints proposed in Giangrande et al. (2013) plays a key role in this aspect; if optimal self-consistency constraints are not provided or do not match theoretical expectations, the precision and accuracy in $K_{DP}$ significantly reduces and larger $window\_len$ values compensate this by oversmoothing $\Phi_{DP}$ (see Appendix

A for results of the performance of phase_proc_lp with very little influence of self-consistency constraints).

To investigate further the effects of $window\_len$ in the performance of phase_proc_lp, Fig. 9(a)-(b) show the NRMSE and normalized bias of $K_{DP}$ estimates with varying $window\_len$ as function of $Z_H$. In agreement with the patterns observed in the scatter plots from Fig. 8, $window\_len$ of 5 produced the best performance compared to other parameter settings. Interestingly,

even in light precipitation (e.g. $Z_H <$30dBZ) smaller values of $window\_len$ produced the best NRMSE metrics, indicating that larger $window\_len$ do not further improve the precision of phase_proc_lp. Instead, larger $window\_len$ enhanced the bias of $K_{DP}$ relative to $K_{DP}^{sc}$ as shown in Fig. 9(b). The parameter $window\_len$ of 5 produced undoubtedly the best metrics for phase_proc_lp and it was selected as the optimal parameter.

### 3.1.4 Wradlib's Vulpiani method

Wradlib's implementation of Vulpiani et al. (2012), kdp_from_phidp, features two optimizing parameters: *winlen* (number of gates used to reconstruct $\Phi_{DP}$) and *dr* (gate length resolution in km). We tested *winlen* values from 3 to 11 and *dr* values from 0.5 to 4. Figure 10 shows multiple scatter plots of $K_{DP}$ estimates using kdp_from_phidp varying settings of $winlen$ and $dr$. Each row of scatter plots holds $winlen$ constant while decreasing $dr$ from left to right. Similarly, each column of scatter plots





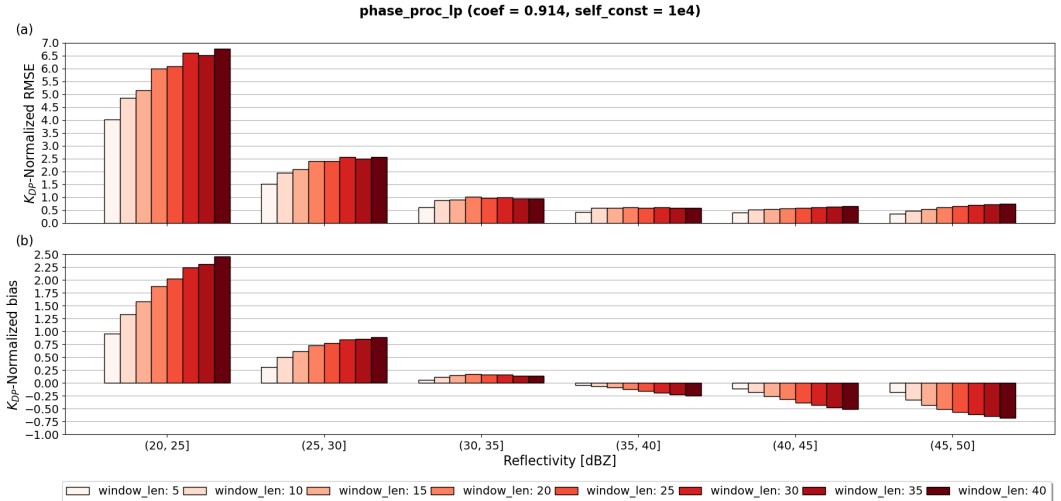

**Figure 9.** Panel (a) shows RMSE normalized by interval-averaged $K_{DP}^{sc}$ of phase_proc_lp relative to $K_{DP}^{sc}$ as a function of reflectivity and for various values of $window\_len$; panel (b) shows same as (a) but for the normalized bias metric.

holds $dr$ constant while decreasing $winlen$ from top to bottom. The first scatter plot, i.e., $winlen$ = 11 and $dr$ = 4, shows $K_{DP}$ clustered predominantly around 0 deg km$^{-1}$ across the entire $Z_H$ range, indicating substantial oversmoothing of $\Phi_{DP}$. Even for $Z_H \geq 30$ dBZ, the noticeable underestimation of $K_{DP}$ relative to $K_{DP}^{sc}$ indicates that kdp_from_phidp is not able to capture signatures of heavy precipitation for large $winlen$ and $dr$ settings. Moving towards the right-most end of the first row of scatter plots, a smaller $dr$ enhances the accuracy of kdp_from_phidp, particularly for $Z_H \geq 30$ dBZ. However, the gain in accuracy comes together with loss in precision in $K_{DP}$ estimates, indicated by the wider spread of data. In addition, decreasing $dr$ makes kdp_from_phidp more prone to the inclusion of outliers, illustrated by data points with $K_{DP} > 1$ deg km$^{-1}$, even for $Z_H \leq 20$ dBZ. The scatter plots from the second row follow the same behavior as in the first row except for wider spread of data, suggesting that decreasing $winlen$ while holding $dr$ constant overall reduces the precision of kdp_from_phidp. When moving from left to right in the second row of scatter plots, accuracy increases while precision decreases when decreasing $dr$. In the last row, $K_{DP}$ estimates are the most scattered for the same $dr$, indicating loss in precision of kdp_from_phidp when reducing $winlen$. The scatter plot of $K_{DP}$ for the smallest parameter settings tested ($winlen$ = 3 and $dr = 0.5$) resembles a scatter plot of random noise with no significant clustering of data, and suggesting extremely poor correlation relative to $K_{DP}^{sc}$. Comparing the scatter plots row-wise and column-wise, decreasing $winlen$ or $dr$ significantly deteriorates the precision of the method. However, the effect on the accuracy is more complex; simultaneous setting of $winlen$ and $dr$ to large values lead to substantial underestimation of $K_{DP}$, whereas small values lead to noisy $K_{DP}$, respectively. These results suggest that the effects of varying $winlen$ and $dr$ in the performance of kdp_from_phidp are strongly intertwined, requiring more analysis in the trade-off between accuracy and precision offered by variations of these parameters.

To analyse the trade-off between accuracy and precision when $winlen$ and $dr$ in kdp_from_phidp, Fig. 11(a)-(b) show the





NRMSE and normalized bias of $K_{DP}$ estimates with varying $winlen$ and $dr$ as function of $Z_H$. Even though Fig. 11(a) has

been clipped at $5.0$, it is important to notice the significantly high values when using the smallest $dr$ ($97.6$, $135.4$ and $278.6$ for $winlen$ of 11, 7, and 3, respectively). The predominantly higher NRMSE values with the smallest $dr$ indicate that the precision of kdp_from_phidp reduces significantly with $dr < 1$ for any $winlen$ tested. An exception occurs in the $Z_H$ interval $(45, 50]$ dBZ, where the smallest $dr$ yield the best metrics due to slight improvements in the accuracy. Despite the limited amount of data within this $Z_H$ interval (see Fig. 3), the clustering of $K_{DP}$ around $K_{DP}^{sc}$ in Fig. 10 and the small normalized

biases in Fig. 11(b) suggest that accuracy improved slightly for the smallest $dr$. The smaller NRMSE with high $dr$ in Fig. 11(a) is counterbalanced by the predominantly larger negative bias for larger $dr$ in Fig. 11(b). This implies that larger $dr$ values in kdp_from_phidp lead to underestimation of $K_{DP}$ for all $winlen$ tested. As a conclusion, combining large $winlen$ with smaller $dr$ produces best performance in heavier precipitation (i.e., $Z_H > 30$ dBZ), whereas combining large $winlen$ with larger $dr$ produces best results for light precipitation. Overall, small values of $winlen$ reduces significantly the precision in the

method without improving accuracy. The parameter setting with the smallest 35-50-dBZ-averaged NRMSE was $winlen = 11$ and $dr = 2$.

### 3.2 Performance Assessment of methods Relative to $K_{DP}^{sc}$

With parameter-optimized kdp_maesaka, kdp_vulpiani, phase_proc_lp and kdp_from_phidp, and including kdp_schneebeli and kdp_iris, we evaluated the relative performance of the methods by quantifying the uncertainties as a function of $Z_H$.

To achieve this, a qualitative analysis with the use of $K_{DP}$ vs. $Z_H$ scatter plots for each method is provided in Sec. 3.2.1. These plots help to evaluate the relative performance of each method against benchmarking $K_{DP}^{sc}$ as a function of $Z_H$. Then, a quantitative assessment of the method's performance is provided in Sec. 3.2.2, by quantifying the uncertainties associated to each method relative to $K_{DP}^{sc}$ using the NRMSE and normalized bias metrics.

### 3.2.1 Qualitative Assessment

We qualitatively assessed the precision and accuracy of the estimated $K_{DP}$ using scatter plots of $K_{DP}$ vs. $Z_H$ for each method. Figure 12 shows six scatter plots comparing the performance of kdp_maesaka, kdp_vulpiani, phase_proc_lp, kdp_from_phidp, kdp_schneebeli and kdp_iris in estimating $K_{DP}$. Each scatter plot illustrate the relationship between estimated $K_{DP}$ (y-axis) relative to $Z_H$ (x-axis), against benchmarking $K_{DP}^{sc}$ (black dashed line). For the parameter-optimized methods in Figs. 12(a)-(d), the optimal parameter selected is indicated in the plot's title together with the method's name. Comparing the scatter

plots, phase_proc_lp demonstrates the highest accuracy and precision, evidenced by the data narrowly clustered around $K_{DP}^{sc}$ across the entire $Z_H$ range. Methods kdp_from_phidp and kdp_schneebeli show the least accuracy and precision, with broader spread and more outliers, particularly when $Z_H < 30$ dBZ. For higher $Z_H$ values, even though kdp_from_phidp shows better precision but worse accuracy than kdp_schneebeli, these two methods strongly underestimate $K_{DP}$, evidenced by the predominant clustering of $K_{DP}$ estimates below $0.5$ deg km$^{-1}$. Method kdp_maesaka shows less scattering of $K_{DP}$ estimates

compared to kdp_from_phidp and kdp_schneebeli, indicating higher precision and accuracy particularly for $Z_H < 30$ dBZ. However, for $Z_H \geq 30$ dBZ, the performance of kdp_maesaka deteriorates rapidly, as shown by the broader spread and sig-

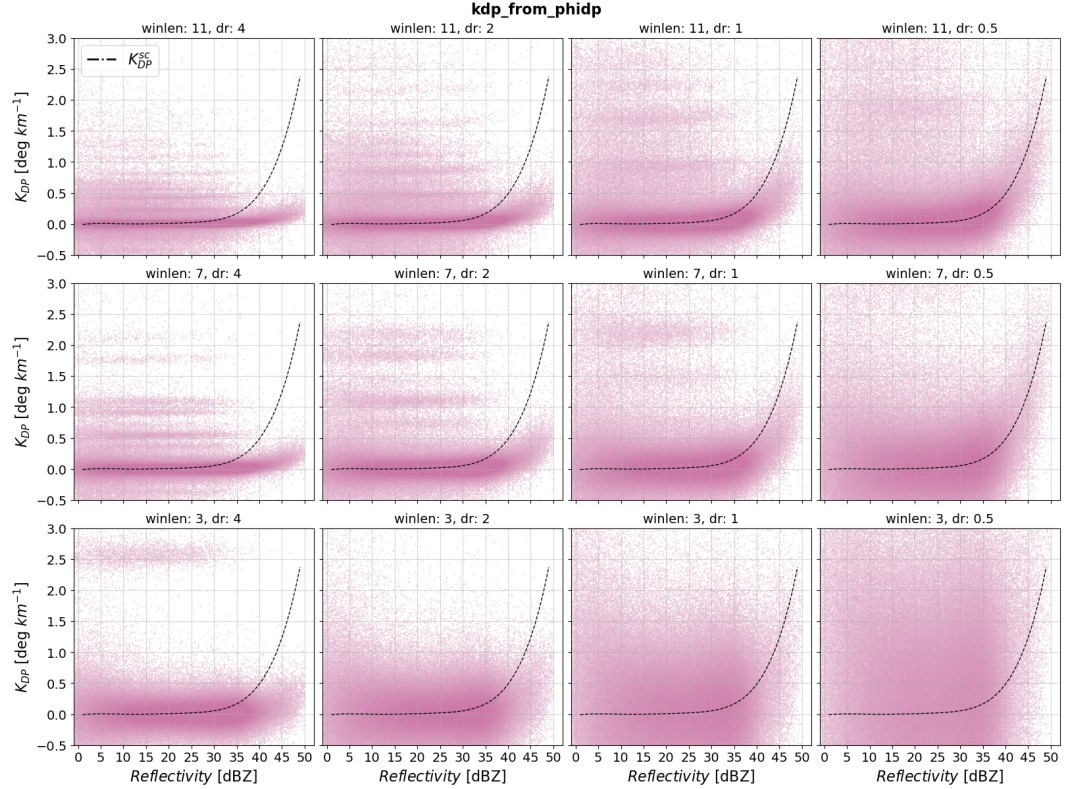

**Figure 10.** Scatter plots of estimated $K_{DP}$ from kdp_from_phidp as function of reflectivity and for various values of $winlen$ and $dr$. Panels (a)-(l) show results with $(winlen, dr)$ tuples values from $(11, 4)$ to $(3, 0.5)$; $winlen$ decreases in intervals of 4 per row whereas $dr$ decreases by half per column. The dashed black line corresponds to $K_{DP}^{sc}$.

nificant underestimation of $K_{DP}$ relative to $K_{DP}^{sc}$. Methods kdp_vulpiani and kdp_iris show moderate performance, with better accuracy and precision than kdp_from_phidp, kdp_schneebeli and kdp_maesaka, but less than phase_proc_lp. Between methods kdp_vulpiani and kdp_iris, kdp_vulpiani shows better correlation of $K_{DP}$ estimates with $K_{DP}^{sc}$ for $Z_H \geq 35$ dBZ, indicating higher accuracy in heavier precipitation. However, kdp_iris shows less scattering across the entire $Z_H$ range, indicating overall higher precision than kdp_vulpiani. Methods kdp_vulpiani, kdp_from_phidp, kdp_schneebeli and kdp_iris include negative $K_{DP}$ values, which should not be expected in rain observations. These negative estimates show up predominantly in lighter precipitation (i.e., $Z_H < 30$ dBZ), indicating they are most likely produced by noise in $\Phi_{DP}$. However, the inclusion of negative $K_{DP}$ estimates are useful, for instance, in the detection of snow crystals, allowing kdp_vulpiani, kdp_from_phidp, kdp_schneebeli and kdp_iris to be used in a wider range of applications when compared to kdp_maesaka and phase_proc_lp. The relatively high accuracy and precision of kdp_iris and kdp_vulpiani, together with the inclusion of negative $K_{DP}$ estimates, leave these two methods as well-suited candidates for QPE, calibration and hydrometeor classification routines.





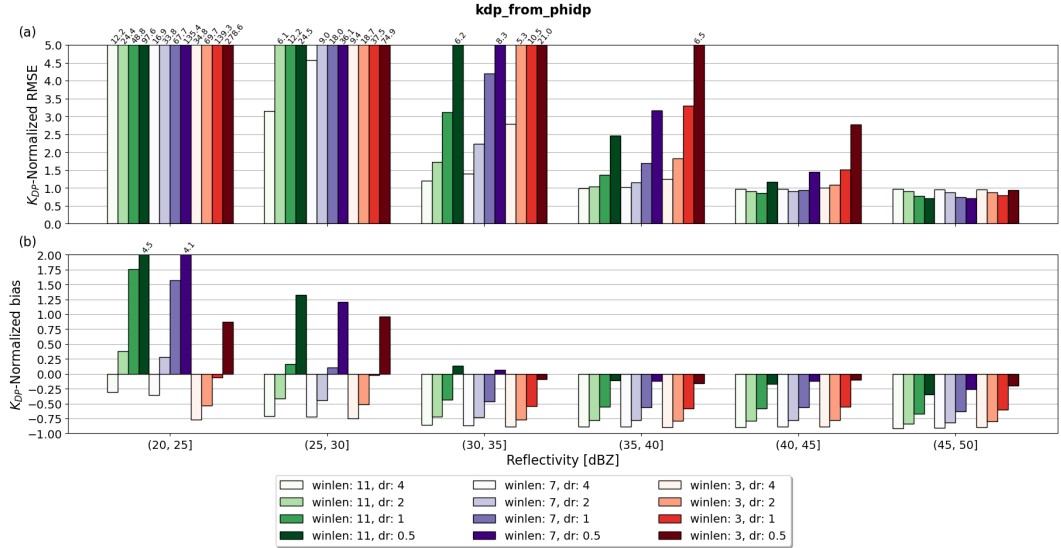

**Figure 11.** Panel (a) shows RMSE normalized by interval-averaged $K_{DP}^{sc}$ of kdp_from_phidp relative to $K_{DP}^{sc}$ as a function of reflectivity and for various values of $winlen$ and $dr$; panel (b) shows same as (a) but for the normalized bias metric. The numbers on top of the bars indicate the values of the metric exceeding the y-axis limit selected

### 3.2.2 Quantitative Assessment

The quantitative assessment of the methods was achieved throuhout the metrics NRMSE and normalized bias, and comple-
mented with statistics from the Wasserstein distance (WD) (Ramdas et al., 2015). The WD measures the similarity between
two cumulative distributions, given in this study by $K_{DP}$ estimated by each method, and $K_{DP}^{sc}$. On the one hand, NRMSE and
normalized bias computed as a function of $Z_H$, allows the assessment of relative accuracy and precision of the methods based
on precipitation intensities. The WD, on the other hand, estimated independently from each radar scan and with statistics over
the entire set of scans, allows the assessment of the relative consistency and robustness of the methods.

Figures 13(a)-(b) show NRMSE and normalized bias of estimated $K_{DP}$ for each method. Overall, phase_proc_lp shows the
best performance, as evidenced by the lowest NRMSE values in Fig. 13(a) and moderately low bias in Fig. 13(b) across all
$Z_H$ intervals. In contrast, kdp_schneebeli shows the worst performance among the methods, indicated by the highest NRMSE
values and moderately high bias across all $Z_H$ intervals. Method kdp_from_phidp shows substantially higher NRMSE values
than kdp_maesaka, kdp_vulpiani, phase_proc_lp and kdp_iris but significantly smaller than kdp_schneebeli, particularly for
the smallest $Z_H$ values. The relatively small bias of kdp_from_phidp when NRMSE values are substantially high, is explained
by the positive-to-negative symmetrical spread of $K_{DP}$ estimates around the x-axis, indicating poor precision. Additionally,
the persistent negative and large normalized bias of this method relative to the other methods indicates kdp_from_phidp un-
derestimates $K_{DP}$ the most. Methods kdp_maesaka, kdp_vulpiani and kdp_iris have moderate NRMSE values, performing





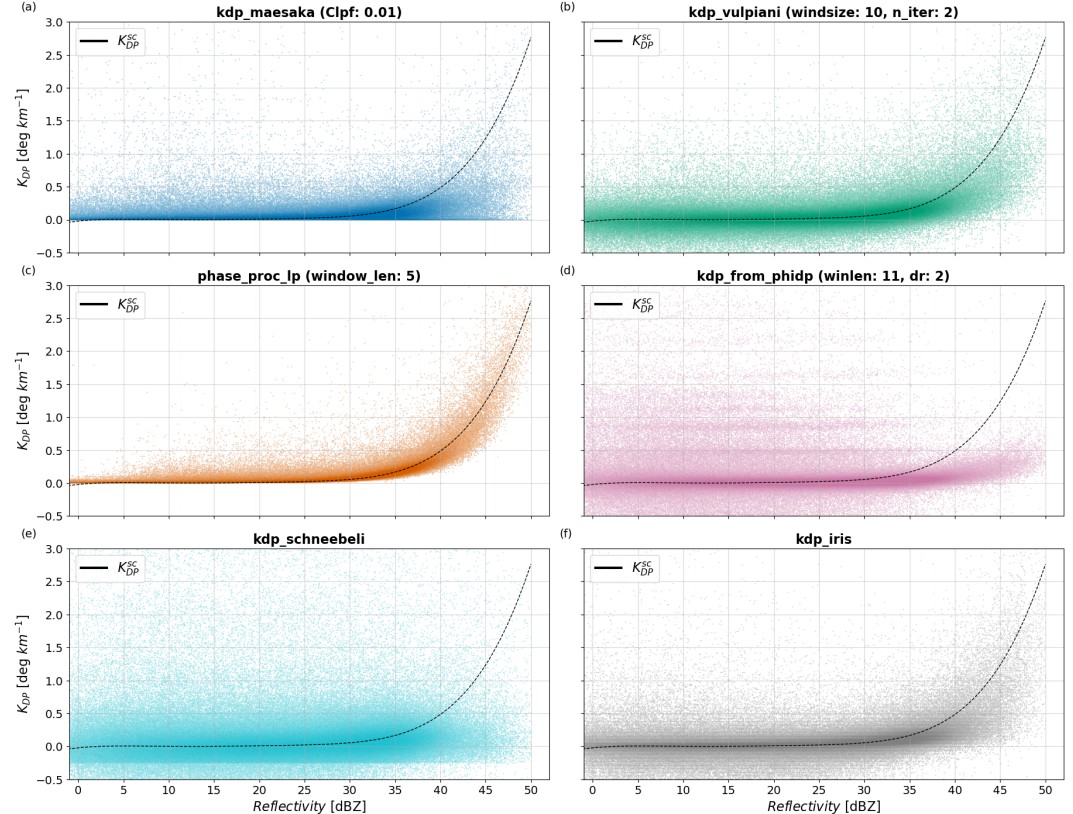

**Figure 12.** Scatter plot of estimated $K_{DP}$ from each parameter-optimized method relative to $K_{DP}^{sc}$ as function of reflectivity. Panels (a)-(f) show kdp_maesaka, kdp_vulpiani, kdp_iris, phase_proc_lp, *kdp_from_phi_dp* and kdp_schneebeli, respectively. The dashed black line corresponds to $K_{DP}^{sc}$.

better than kdp_schneebeli and kdp_from_phidp but not as well as phase_proc_lp. Among these three methods, kdp_maesaka has the smallest NRMSE values for $Z_H \leq 35$ dBZ but the largest when $Z_H \geq 40$ dBZ. The relatively large positive bias of kdp_maesaka when $Z_H < 30$ is a direct consequence of the exclusion of negative $K_{DP}$ estimates. However, the persistent larger negative bias of kdp_maesaka relative to kdp_vulpiani and kdp_iris when $Z_H \geq 30$ dBZ, indicates stronger underestimation of $K_{DP}$ and thus lesser accuracy. These results indicate that, in comparison to others, kdp_maesaka performs slightly

better in light precipitation (i.e., $Z_H < 30$ dBZ) but worse in heavier precipitation. Between kdp_vulpiani and kdp_iris, kdp_iris shows overall smaller NRMSE and normalized bias, indicating higher accuracy and precision than kdp_vulpiani.

  Complementary to NRMSE and normalized bias metrics, we evaluated the consistency and robustness of the methods using the Wasserstein distance (WD). The WD was computed for each radar scan independently using the *wasserstein_distance* module from SciPy (Virtanen et al., 2020). Then, the statistics from the estimated WD values for all scans were visualized

and analysed using using boxplots. Figure 14 consists of two panels comparing the WD boxplots of the methods. Figure 14(a) compares the WD for all methods, including kdp_schneebeli, which presented significantly large WD. Figure 14(b) presents





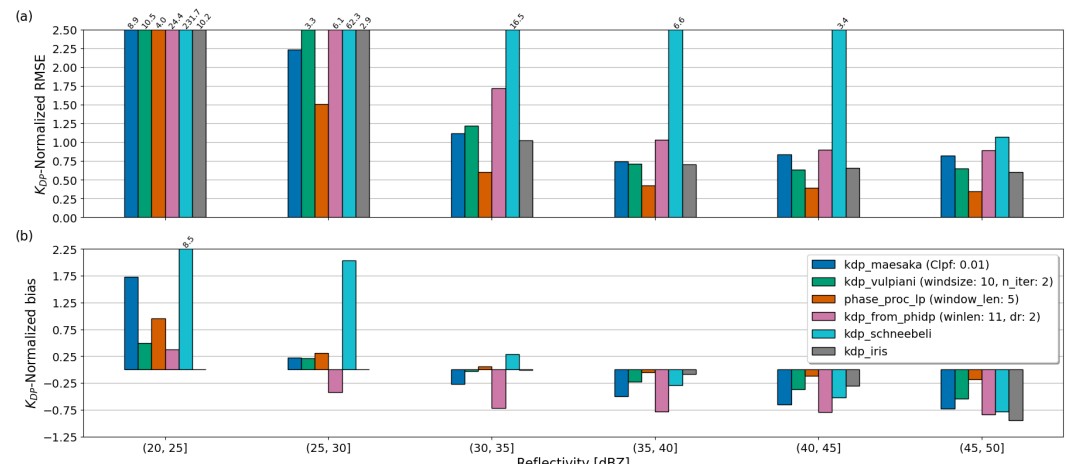

**Figure 13.** Panel (a) shows bias of estimated $K_{DP}$ from each parameter-optimized method relative to $K_{DP}^{sc}$ as function of reflectivity; panel (b) shows same as (a) but bias normalized by interval-averaged $K_{DP}^{sc}$. The numbers on top of the bars indicate the values of the metric exceeding the y-axis limit selected

the same data as (a) but excluding kdp_schneebeli to better compare the remaining methods. Each boxplot summarizes the statistics of estimated WD by showing the median (black dashed line), interquartile ranges (IQR), $1.5\times$ the IQR (whiskers) and outliers (crosses). The insights provided by boxplots in this analysis are twofold. First, a WD median closer to 0 indicates

higher similarity between the cumulative distributions of a method's estimated $K_{DP}$ and that from $K_{DP}^{sc}$, ultimately indicating higher accuracy. Second, a narrower IQR indicates less variability of a method's performance between scans, indicating higher consistency.

In Figure 14(a), the x-axis lists six methods: kdp_maesaka, kdp_vulpiani, phase_proc_lp, kdp_from_phidp, kdp_schneebeli

and kdp_iris. The y-axis measures the WD values, ranging from 0 to 2. The boxplot for method kdp_schneebeli shows the largest WD with a median of 0.33, an IQR from 0.18 to 0.45, and several outliers. The other methods (kdp_maesaka, kdp_vulpiani, phase_proc_lp, kdp_from_phidp and kdp_iris) have median WD values ranging from 0.0 to 0.1, with smaller IQRs and fewer outliers. In Figure 14(b), method kdp_schneebeli is excluded, allowing for a clearer comparison of methods kdp_maesaka, kdp_vulpiani, phase_proc_lp, kdp_from_phidp and kdp_iris. The y-axis is rescaled to range from 0.0 to 0.2 for

better visualization. Method phase_proc_lp has the lowest WD median at 0.01 with a narrow IQR from 0.008 to 0.018. Method kdp_from_phidp has significantly larger WD median of 0.098 and IQR from 0.077 to 0.122. Methods kdp_maesaka and kdp_iris have WD medians of 0.026 and 0.041, respectively, with moderate IQR and few outliers. Method kdp_vulpiani has a moderate WD median of 0.049 but noticeably wider IQR from 0.033 to 0.096 when compared to kdp_maesaka, phase_proc_lp, kdp_from_phidp and kdp_iris.






The large WD median of kdp_schneebeli indicates it performs worse compared to the other methods, overshadowing the performance difference among the remaining methods. Additionally, the large IQR of kdp_schneebeli implies that the method does not perform consistently thus reducing its reliability. Method phase_proc_lp demonstrates the best and consistent performance with the lowest WD median and narrowest IQR. These results additionally indicate that the distribution of $K_{DP}$ estimated from phase_proc_lp is the closest to $K_{DP}^{sc}$. It is important to remember here that phase_proc_lp is supported by self-consistency relations constraining $K_{DP}$ estimates based on $Z_H$ observations, ultimately enhancing it's accuracy and stability. The moderate IQR and significantly larger WD median of kdp_from_phidp, indicate it's performance is consistent albeit less accurate relative to the other methods. Method kdp_vulpiani in turn, has a moderate WD median but relatively larger IQR, indicating better accuracy than kdp_from_phidp although less consistent. Methods kdp_maesaka and kdp_iris show similar consistency and accuracy, evidenced by their relatively low WD medians and moderate IQRs. These findings suggest that while kdp_schneebeli is the less accurate and consistent, the performance among the remaining methods vary, with phase_proc_lp presenting the highest robustness provided the method with quality-controlled $Z_H$ and optimized self-consistency settings.

### 3.3 Consistency Analysis of $K_{DP}$ Retrievals from methods

Each method has its unique combination of mathematical approach, data requirements, and constraints (see Table 1), indicating uniqueness in the $K_{DP}$ fields produced. How similar or dissimilar are these outputs is not clearly visible from the metrics computed nor the scatter plots displayed in Sec. 3.2. To answer this question, we study the consistency among methods using $K_{DP}$ vs $K_{DP}$ correlation plots shown in Fig. 15. Each scatter plot in Fig. 15 shows the relationship between $K_{DP}$ estimated by a method (y-axis) with respect to $K_{DP}$ estimated from a different method (x-axis) and the Pearson correlation coefficient ($R$) is shown in the upper left corner of each scatter plot. The axes range from $-0.5$ to $3.0$ deg km$^{-1}$ to include negative $K_{DP}$ estimates. This part of the analysis does not require any ground-truth framework, allowing the use the entirety of radar dataset, i.e., including the attenuated observations (see red data in Fig. 3).

In Figure 15, the scatter plot of kdp_iris against kdp_vulpiani shows the best correlation among the methods, illustrated by the data significantly clustered along the diagonal and corroborated by the highest $R$ of $0.66$. Methods kdp_iris and kdp_vulpiani correlate similarly to phase_proc_lp, indicated by the second highest $R$ of $0.65$ for both. In relation to kdp_maesaka, the consistencies of kdp_iris and kdp_vulpiani are rather moderate, whereas in relation to kdp_from_phidp and kdp_schneebeli, they are significantly poorer. Among the methods, kdp_schneebeli correlates the least with any of the methods, evidenced by the data widely spread along the axes and showing negligible clustering of data along the diagonal. Particularly kdp_schneebeli against kdp_from_phidp shows the worst consistency with $R = 0$ and the majority of data clustered around the x- and y-axis. Method phase_proc_lp correlates moderately to kdp_maesaka with an $R = 0.41$, although the scatter plot does not exhibit any particular pattern or clustering of data along the diagonal. Relative to kdp_from_phidp, phase_proc_lp shows significantly lower $R$ despite the clear data correlation off of the diagonal. However, the small $R$ value becomes evident when observing the dense clustering of data around $0$ deg km$^{-1}$ for phase_proc_lp. This results indicates the consistency between kdp_from_phidp and phase_proc_lp is highly influenced by the negative $K_{DP}$ estimates in kdp_from_phidp that are mapped to $0$ deg km$^{-1}$ in





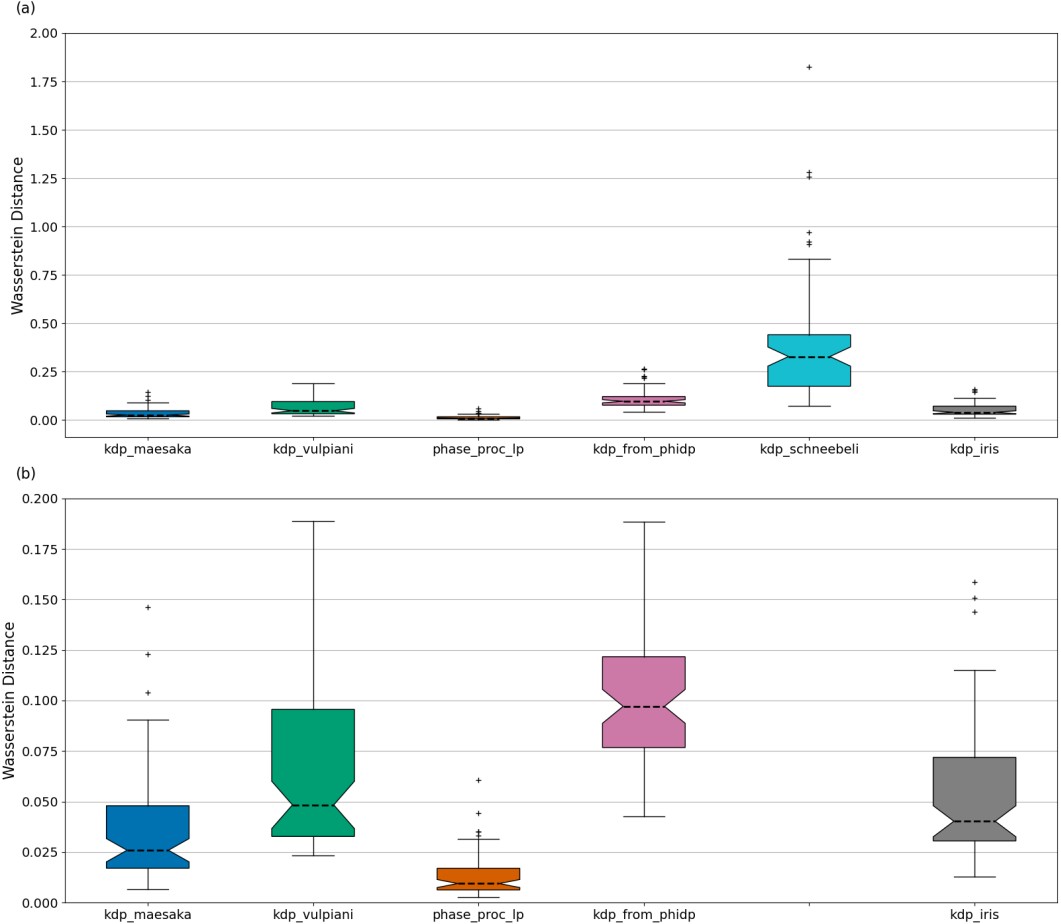

**Figure 14.** Panel (a) shows the boxplot of computed WD for each parameter-optimized method; panel (b) shows same as (a) but excluding kdp_schneebeli for better visualization of the outperforming methods. The boxplot displays the WD median (black dashed line), IQRs (boundaries of the box), $1.5\times$ the IQR (whiskers) and the outliers (black crosses).

phase_proc_lp. Overall, the scatter plots show that kdp_from_phidp underestimates $K_{DP}$ relative to the other methods. Method kdp_maesaka shows no significant correlation with any method, with the largest $R$ being $0.41$ relative to both phase_proc_lp and kdp_vulpiani.

## 4  Conclusions

In this study, we conducted a comprehensive evaluation of several $K_{DP}$ estimation methods using C-band weather radar data, with a focus on their performance in rainfall observations. We employed a self-consistency framework, that links $Z_H$ and $Z_{dr}$ observations with $K_{DP}$, as the basis for our evaluations. This approach allows for the construction of the reference $K_{DP}$ observations that can be used to assess the accuracy and robustness of the studied $K_{DP}$ estimation methods. The use of the





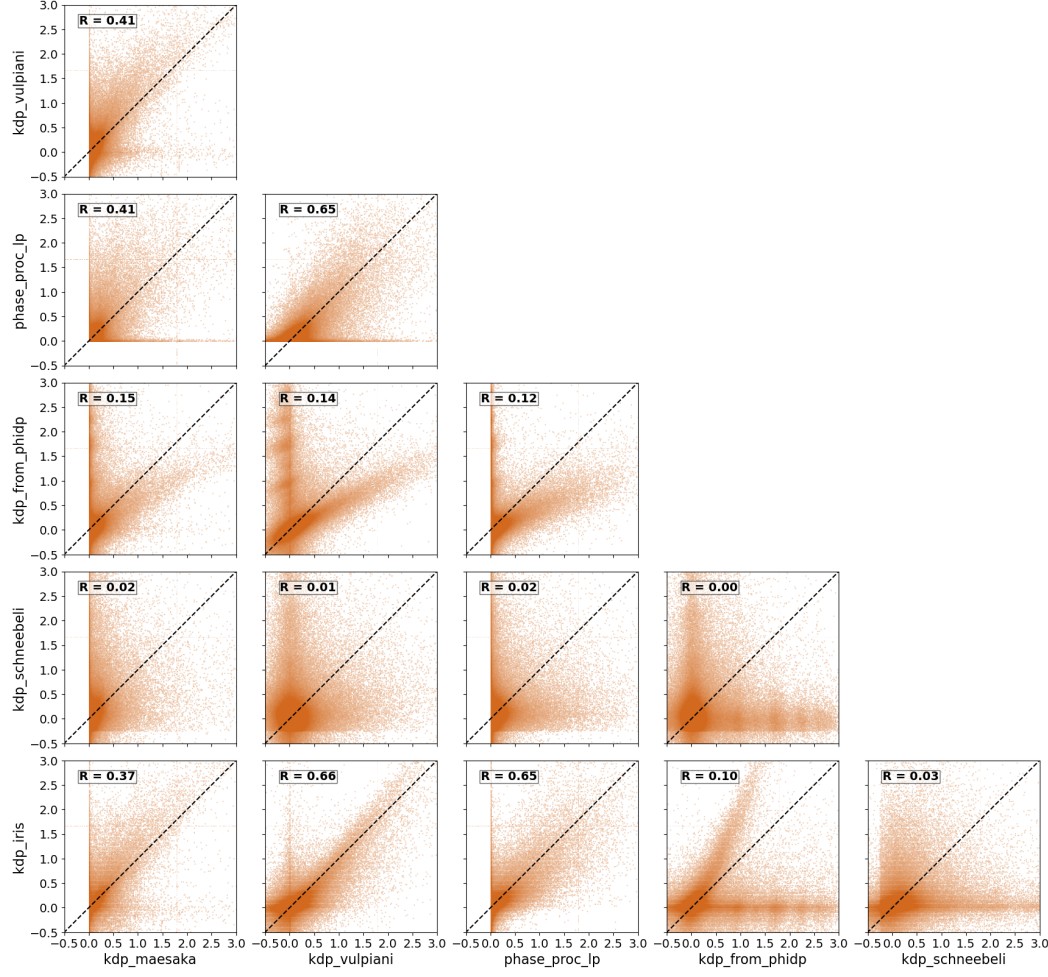

**Figure 15.** Correlation plot between the methods. Each scatter plot shows the relationship between two different methods without repetition and no method is compared to itself. The x- and y-axis represent $K_{DP}$ estimated by a method in units of deg km$^{-1}$. Each plot shows the Pearson correlation coefficient between the two compared methods.

self-consistency framework requires rather strict quality control which is described in the paper.

Some, four out of six, of the $K_{DP}$ estimation methods have user-configurable parameters. Using the proposed evaluation framework we could define optimized parameter settings. Most of the methods showed significant improvement in the performance after the optimization.

By comparing the relative performances of the estimation methods over the range of rain intensities, as characterized by

the radar $Z_H$ values, we have found significant difference in performances of the evaluated methods. Overall, implementations





of Giangrande et al. (2013), Vulpiani et al. (2012) and Wang and Chandrasekar (2009) exhibited the lowest NRMSE and normalized biases over the studied range, from 20 to 50 dBZ of $Z_H$ values.

Our comparative analysis revealed that while the implementation of Giangrande et al. (2013)'s method stands out for its high accuracy and precision, its performance is heavily dependent on the provided self-consistency constraint. Without proper optimization of the self-consistency relation, linking $Z_H$ and $K_{DP}$, and quality control of $Z_H$, even the best window length setting for this method can lead to suboptimal results, i.e. higher RMSE and $K_{DP}$ underestimation at higher $Z_H$ values. It should be noted, however, since the reference framework and Giangrande et al. (2013)'s method both use consistency relations $K_{DP}(Z_H Z_{dr})$ and $K_{DP}(Z_H$, respectively, they are not independent. Therefore, it is possible that the part of reported perfor-
mance is caused by this dependence. Implementations of Vulpiani et al. (2012) and Wang and Chandrasekar (2009) showed good performance and do not require use of other radar variables, which potentially make them less sensitive to radar data quality issues, such as calibration and attenuation.

An additional qualitative comparison of the methods performances was carried out by computing correlations of derived $K_{DP}$
values from the dataset that also included attenuated radar observations. The correlation between $K_{DP}$ values estimated using different methods is not very high. The highest correlation values of 0.65-0.66 were observed between Giangrande et al. (2013), Vulpiani et al. (2012) and Wang and Chandrasekar (2009) methods. This indicates that uncertainty between different precipitation estimates could stem from the differences in the used $K_{DP}$ methods.

The study is based on the self-consistency framework that limits it to the cases where no significant attenuation is observed. Additionally, the scope of our study is limited to the Finnish climatology and a single radar frequency, namely C-band radar observations. Despite these limitations, our findings offer valuable guidance for the use of $K_{DP}$ estimation methods in rainfall observations. These results have significant implications for both operational radar network and hydrometeorological research, where the accuracy, precision, and stability of $K_{DP}$ estimates are crucial.

**Appendix A:  Influence of self-consistency constraint in *phase_proc_lp***

Figure A1 shows same scatter plots as in 8, with $K_{DP}$ estimated from phase_proc_lp using $self\_const$ of $10^6$ instead of $10^4$. The motivation behind was to study the performance of phase_proc_lp with little influence of self-consistency constraints. In Giangrande et al. (2013), the non-negativity condition in $K_{DP}$ estimates is ensured by restricting the b-vectors: $\boldsymbol{b} \geq 0$. In addition, to produce more realistic $K_{DP}$ estimates, they introduced the self-consistency relation $K_{DP}(Z_H) = aZ_H^b$ to bound
the estimates based on observed $Z_H$, requiring that the user provides quality controlled data. The restriction of the b-vectors becomes $\boldsymbol{b} \geq aZ_H^b$, which in phase_proc_lp is implemented as $\boldsymbol{b} \geq (10^{0.1 \times Z_H})^{coef}/self\_const$. Therefore, a two order of magnitudes larger $self\_const$ value was used in this study to test the performance of phase_proc_lp with a significantly reduced influence of self-consistency constraints. The scatter plots show $K_{DP}$ data clustered around $K_{DP}^{sc}$ up to 35 dBZ. Beyond





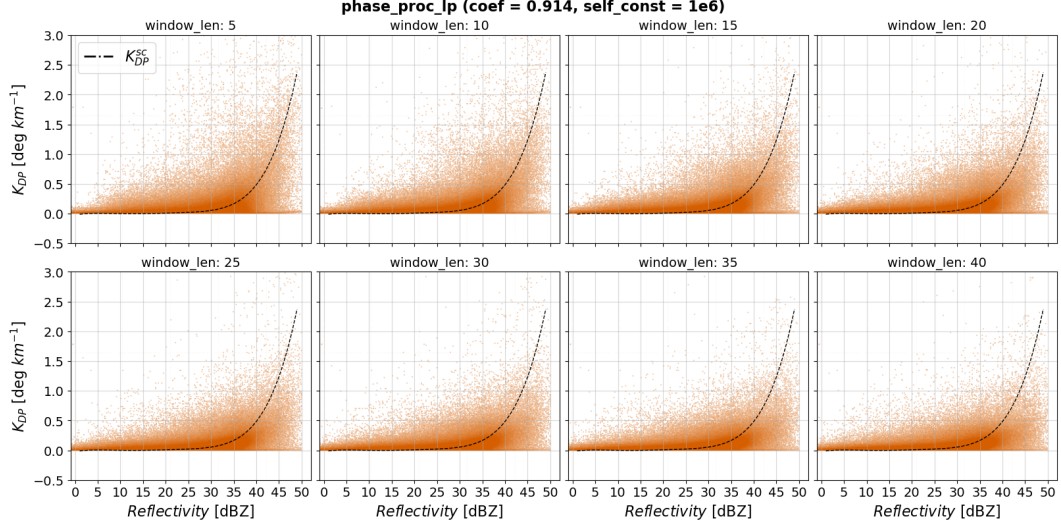

**Figure A1.** Scatter plots of estimated $K_{DP}$ from phase_proc_lp as function of reflectivity and for various values of $window\_len$. Panels (a)-(h) show results with $window\_len$ values from 5 to 40, while fixing $coef$ to 0.914 and $self\_const$ to $10^6$. The solid black line denotes corresponds to $K_{DP}^{sc}$.

this threshold, precision and accuracy decays significantly regardless of the window length. However, in scatter plots with larger

window lengths $K_{DP}$ data is less scattered across the entire $Z_H$ range and only slightly less accurate after 35 dBZ.

To further investigate the effects of the self-consistency constraint in phase_proc_lp, Fig. A2(a)-(b) show the normalized RMSE and bias of $K_{DP}$ (estimated with $self\_const = 10^6$) relative to $K_{DP}^{sc}$. Interestingly, the normalized RMSE in Fig. A2(a) behaves inversely as in normalized RMSE in Fig. 9, whereas normalized bias shows similar behavior for both. The opposite

behaviors in normalized RMSE results indicate that window length has a strong impact in the performance of phase_proc_lp depending on whether the adequate self-consistency settings were provided; if so, smaller window lengths yield better performance by capturing fine scale precipitation features specially in heavy precipitation. In the opposite case, larger window lengths yield better performance by oversmoothing $\Phi_{DP}$ thus reducing the impact of noise at the expense of loosing fine-scale precipitation features. The oversmoothing effect from larger window lengths in $K_{DP}$ is also implied from the normalized bias shown

in Fig. A2(b); larger window lengths produced absolute largest biases in both extremes of the $Z_H$ range. In addition, even though the normalized bias shows similar behavior for $self\_const = 10^6$ and $self\_const = 10^4$, the latter produces larger differences between window lengths, indicating that high accuracy and precision of phase_proc_lp predominates in smaller window lengths, provided the adequate self-consistency constraints and quality-controlled $Z_H$.



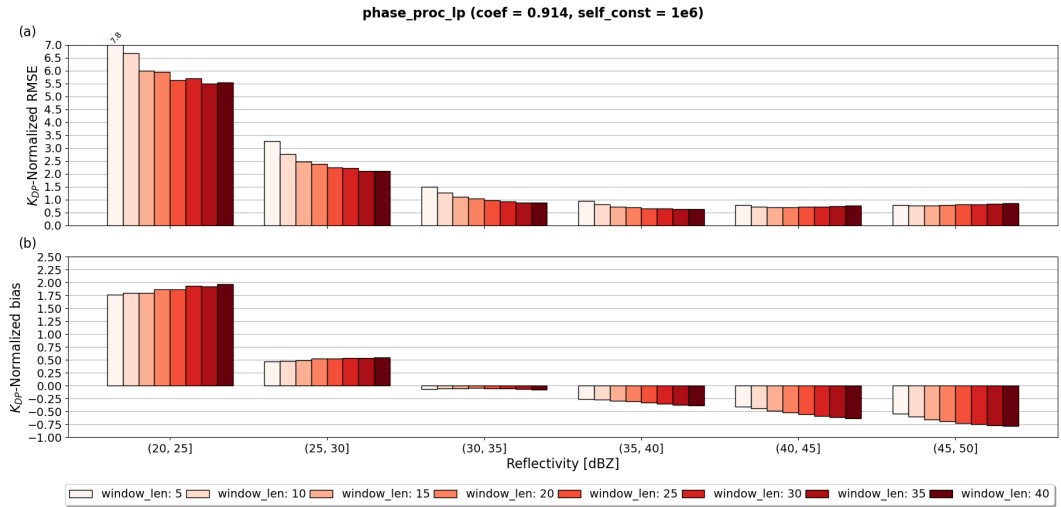

**Figure A2.** Panel (a) shows RMSE normalized by interval-averaged $K_{DP}^{sc}$ of phase_proc_lp relative to $K_{DP}^{sc}$ as a function of reflectivity and for various values of $window\_len$; panel (b) shows same as (a) but for the normalized bias metric.



**Data Availability**

The radar raw data and KDP dataset, accessed via the link in Aldana (2024), includes the raw radar data and KDP processed data used to analyse the KDP estimation methods. The data has been processed using Python and it includes:

– The folder "radar" includes several subfolders: "yyyy/mm/dd/iris/raw/VAN". The "VAN" subfolder includes the .raw radar with PPI's observed by Vantaa radar at an elevation angle of 0.7 for a specific time: "yyyymmddHHMM_VAN.PPI3_B.raw". This data can be further read with PyArt (Helmus and Collis, 2016).

– The folder "KDP_data" includes 5 .hdf5 files storing tables containing information about date (in pandas numerical value. It requires transformation to datetime object), Z (in dBZ), Zdr (in dB), attenuated gate (as boolean), theoretical or self-consistency KDP (in deg / km) and computed KDP (in deg / km) from a given method for different settings. The method is indicated in the name of the file as kdp_method_scatter.hdf5, where method can be:

– 'iris_sch', referring to table containing KDP from iris software (used in the Finnish Meteorological Institute) and
KDP computed from PyArt's implementation of Schneebeli et al. (2014). These two methods were computed together because only one KDP output was retrieved. They do not feature any user-configurable parameters to test.

– 'mae', referring to table containing KDP computed from PyArt's implementation of Maesaka et al. (2012). The columns correspond to KDP computed by varying parameter 'Clpf'.

– 'vulpiani', referring to table containing KDP computed from PyArt's implementation of Vulpiani et al. (2012). The
columns correspond to KDP computed by varying parameters 'windsize' and 'n_iter'.

– 'pplp', referring to table containing KDP computed from PyArt's implementation of Giangrande et al. (2013). The columns correspond to KDP computed by varying parameter 'windowlen'.

– 'wradlib', referring to table containing KDP computed from Wradlib's implementation of Vulpiani et al. (2012). The columns correspond to KDP computed by varying parameters 'winlen' and 'dr'.

The disdrometer dataset to obtain the DSD parameters and be accessed via the link provided in Moisseev (2024).

**Author Contributions**

MA conducted the investigation process, collected the data, performed the formal analysis of the data and visualization; MA, SP and DM designed the methodology; SP, AL, MK and DM formulated the research goals and aims; AL and DM provided data; MA prepared the manuscript draft; MA, SP, AL, MK and DM reviewed, commented and edited the manuscript.

**Competing Interests**

The contact author has declared that none of the authors has any competing interests



**Acknowledgements**

We thank Jenna Ritvanen for her valuable comments for improving the visualization of the data; we thank colleagues from the Early Career Community at the Finnish Meteorological Institute for feedback on how to make the manuscript reachable also for general audience.





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
