# Peer review of "Benchmarking $K_{DP}$ in Rainfall: a Quantitative Assessment of Estimation Algorithms Using C-Band Weather Radar Observations"

_Atmospheric Measurement Techniques, 2024_

## Author Response (AR1)

**Response to Community Comments & Reviewers**

We would like to thank the reviewers for their thoughtful and constructive feedback. We have carefully addressed each of the points raised and we have updated the manuscript accordingly. Below, we provide detailed responses to each of the comments.

**Community Comments**

**CC1 (**https://doi.org/10.5194/amt-2024-155-CC1**)**

1. This is a well prepared paper with interesting and useful results. There are many references to relevant publications on the topic. There is one paper that I believe should be included because it was the first paper that provided relationships between C-band polarimetric radar observables in rainfall, including KDP:

   C-Band Dual-Polarization Radar Observables in Rain
   K. Aydin and V. Giridhar
   Print Publication: 01 Aug 1992
   DOI: https://doi.org/10.1175/1520-0426(1992)009<0383:CBDPRO>2.0.CO;2 Page(s): 383–390

**Author's response:**

1. We found the proposed publication pertinent to the current manuscript, particularly when introducing and highlighting the use of self-consistency relations among the polarimetric variables.

**Changes in Manuscript:**

1. The paper "-Band Dual-Polarization Radar Observables in Rain" by K. Aydin and V. Giridhar, was added and cited in the Data & Methods Section, line 96, when introducing the self-consistency relations.

**Reviewer Comments**

**RC1 (**https://doi.org/10.5194/amt-2024-155-RC1)

This article discusses an attempt to objectively assess the performance of KDP retrievals in rain. The authors make use of the self-consistency in rain of Zh, Zdr and KDP to retrieve KDP from Zh and Zdr observations from a heavily quality-controlled dataset. They then proceed to compare the self-consistency KDP with optimized KDP retrieval algorithms available from open-source libraries Py-ART and wradlib and the KDP retrieval available in the IRIS software.

It is undoubtedly an interesting and worthy exercise. The article is well written, and the topic is of interest. I have some general comments that I would like that they are addressed before it being accepted for publication.

**General comments:**

1. Many Met Services (including WSR-88D) use a simpler KDP retrieval consisting of one or two moving median windows to smooth PhiDP and a least square retrieval of the KDP. The MeteoSwiss version of Py-ART has functions kdp_leastsquare_single_window, kdp_leastsquare_double_window, smooth_phidp_single_window and smooth_phidp_double_window available. I suggest the authors add those into the evaluated methods. It might be of interest for many Met Services to see where they retrievals stand.

2. I do not understand why the authors did not try to tune KDP Schneebeli and KDP iris. KDP Schneebeli has some parameters that can be tuned (Measurement Error Covariance Matrix, Scaled State Transition Error Covariance Matrix and all the arguments for the pre-filtering of the PsiDP (which is not performed in the article). I am not familiar with the IRIS implementation, but I would assume there are also parameters that can be tuned. Not tuning those algorithms while tuning the others puts them in an unfair position when performing the evaluation.

3. The conclusions should emphasize more the limits of this study. With all the data filtering performed the authors are evaluating KDP in effectively idealized conditions. In operational conditions, robustness to outliers, performance in low SNR, performance in the presence of residual clutter and/or non-liquid precipitation are also important factors. It would be interesting also to have some data on the computational time of each of the retrievals.

**Specific comments:**

4. According to the GitHub repository the right way to write the name of the software package is Py-ART**.**

5. For the sake of completeness, and since they have such a long list of citations, the authors maybe should cite Figueras i Ventura and Tabary (2013) since it was one of the first papers

demonstrating the superior performance of KDP-based algorithms at all 3 regularly used weather radar bands (S, C and X) in an operational context in Europe.

6. A recapitulative table with the parameters chosen for each tunned algorithm would be appreciated. It will also be helpful to highlight the effective range resolution achieved after tunning the retrievals.

**References**

2013: The new French operational polarimetric radar rainfall rate product, J Figueras i Ventura, P Tabary, Journal of Applied Meteorology and Climatology 52 (8), 1817-1835

https://github.com/meteoswiss/pyart

**Author's response:**

We would like to thank you for the thoughtful and constructive feedback. We have carefully addressed each of the points raised and we have updated the manuscript accordingly.

1.  Thank you for suggesting additional publicly available KDP retrieval implementations. We are currently evaluating methods that directly output KDP estimates and therefore only kdp_leastsquare_single_window and kdp_leastsquare_double_window fit within the scope of the article. However, adding two new methods with a total of three parameters to optimize *(wind_len, lwind_len* and *swind_len),* will require new analysis and substantial modification of the current article. Instead, we propose to show the results in this document, hopefully answering the reviewer's question:

[Figure]

[Figure]

**kdp_leastsquare_single_window**

**kdp_leastsquare_double_window**

[Figure]

[Figure]

(a)

(b)

Summary of the findings:

a. kdp_leastsquare_sinle_window optimal parameter was *wind_len* = 11. It was selected from the minimum NRMSE in the range between 40 and 50 dBZ. The range 35 to 40 dBZ was excluded for the selection, since the results become strongly influenced by the small NRMSE for *wind_len* values of 3 and 5, which produced unrealistic results.

b. Kdp_leastsqaure_double_window optimal parameters were l*wind_len* = 41 and s*wind_len* = 11, for the range between 35 and 50 dBZ. This method shows a clear discontinuity at 35 dBZ, in which the long window length is replaced by the short window length. The reflectivity threshold was selected to be 35 dBZ to focus on heavier precipitation.

c. Optimized kdp_leastsquare_double_window produce slightly better results than kdp_leastsquare_sinle_window, most likely to the sensitive to fine scale precipitation features in the heavier precipitation region (>=35 dBZ) by switching to a small smoothing window size. Additionally, the larger window when Z < 35 dBZ, helps to reduce the impact of outliers and noise in the retrieval of KDP.

d. Optimized kdp_leastsquare_double_window and kdp_leastsquare_sinle_window produced better results than kdp_schneebeli but no better than any of other algorithms.

2. We tested parameters related to smoothing window lengths and that do not require any previous knowledge or assumptions of the rainfall fields observed. Regarding kdp_schneebeli, the parameters *rcov* and *pcov* are the error covariance matrices of the measurements and state transitions, respectively, and these require a complex framework to be optimized. Same as the self-consistency relations, they will be unique for a given climatological drop size distribution and radar characteristics. In Schneebeli et al.'s these matrices are computed from a large ensemble of stochastically simulated rainfall fields, providing robust spatial information of KDP and delta_hv (or DSD parameters). Unfortunately, such information is not available to us. The *prefilter_psidp* argument in the method is set to True so that kdp_schneebeli filters the PhiDP hoping to improve the KDP estimates. The method then uses PhiDP as PsiDP and performs the filtering. Regarding kdp_iris, it corresponds to the already implemented method of Wang & Chandrasekar (2009) by Vaisala in the IRIS software. This means that at the FMI end, we receive the KDP output as it comes from IRIS, without us exerting any active role in its computation.

3. We agree that strict data quality control yields highly idealized data. However, it is important to note that our evaluation framework is based on self-consistency of polarimetric variables in rain, which requires such strict data quality control. The self-consistency approach would lose its purpose if applied over nonfiltered data. We will emphasize in the conclusions section that the results of our study are based on highly idealized rainfall data.

4. Thank you for the clarification on how Py-ART is properly written.

5. Thank you for the suggestion of the paper and it has been included when referencing QPE estimated from KDP.

6. A recapitulative table with the parameters tested for each method, including the range of settings tested, the default settings and the optimal settings.

**Changes in Manuscript:**

1. No changes

2. In section 3.1 (Parameter Optimization of methods), we have added in the lines 204-211 an indication to why kdp_schneebeli and kdp_iris are not included in the optimization.

3. In lines 472-474 of the Conclusions Section, we emphasized the high idealized rainfall data used in this study and required for the self-consistency.

4. The name Py-ART was corrected in the entire document.

5. The citation has been included in line 23 inside the Introduction section, when introducing the relevance of KDP in QPE.

6. A recapitulative table has been added in line 350, at the end of the Parameter Optimization of Methods' Section.

**RC2 (**https://doi.org/10.5194/amt-2024-155-RC2**)**

Summary of paper:

The manuscript examines the performance of several commonly-used methods for KDP retrievals in rainfall observations with a C-band weather radar, which is located near Helsinki, Finland. The authors firstly determined carefully quality-controlling radar data by applying several filters to the measured ZH and Zdr and, then, deployed a polarimetric self-consistency approach to compute a reference KDP from radar quantities in rainfall observations. The self-consistency KDP was compared with the derived KDP applying different methods available in Py-ART and wradlib which were optimized accordingly under different conditions. With the performance assessment of different methods, the authors finally directly compared the optimized KDP and calculated their correlation coefficients, which, however indicates generally a high-level difference between different methods.

As general comments, I think the work is very important and comprehensive for the relevant studies in the radar community. It is well written and certainly worthy for publication.

I would like to recommend to add a short introduction to the self-consistency method that is used to probably improve the accuracy of different variables against each other for given hydrometeor types. For the direct comparison between the estimated KDP with different methods, a significance test should be given.

Here are the lists of typos and suggestions (but not limited to):

1. Lines 17-18, to my knowledge, Höller et al., (1994) developed one of the first algorithms for hydrometeor classification by using ZDR, LDR, KDP and phiHV measurements during the evolution of a thunderstorm and the authors should probably cite it.

Reference: Höller, H., Hagen, M., Meischner, P. F., Bringi, V. N., and Hubbert, J.: Life Cycle and Precipitation Formation in a Hybrid-Type Hailstorm Revealed by Polarimetric and Doppler Radar Measurements, J. Atmos. Sci., 51, 2500– 2522, https://doi.org/10.1175/1520-0469(1994)051<_x0032_500:LCAPFI>2.0.CO;2, 1994.

2. Lines 102-103, is Thurai et al. (2007) one of the settings? Please check it.
3. Lines 109-110, please rephrase the sentence "Observations were … only."
4. Figure 4 (probably also other figures), Please add numbers (or letters) to the labels of figure.
5. Lines 233-234, please rephrase the sentence "whereas …"
6. Line 256, kDP -> KDP

7. Figure 6, which panel is (a)-(p)? Please add.
8. Lines 343-348, is this paragraph necessary? Please make a check.
9. Line 400, remove one of the "using".
10. Line 426, it's -> its.
11. Line 427, it's -> its.
12. Line 433, remove "from methods"
13. Line 456, a significance test should be given when the comparisons between two KDP are conducted.
14. Figure 15, remove "plot".
15. Lines 478-479, please rephrase the sentence "both use …… independent."
16. Line 496, in 8 -> in Figure 8 (I think).
17. Lines 669-670, it seems the reference is not complete. Please revise it.

**Author's response:**

We would like to thank you for the thoughtful and constructive feedback. We have carefully addressed each of the points raised and we have updated the manuscript accordingly.

We agree that an introduction to the self-consistency approach was missing, and we have added a substantial paragraph to the introduction dedicated to it. Regarding the significance test, the assessment methodology adopted, and the scope of the analysis do not lend themselves to the formulation of a meaningful hypothesis or the conditions required for statistical significance testing. As a result, we believe that adding this test would not enhance the value of the findings or contribute additional insights.

1. Thank you for the suggestion of an earlier publication relevant to hydrometeor classification using KDP. It has been included.
2. Those correspond to the settings used in the T-Matrix calculations and not for the KDP methods. For clarity, we changed the words used.
3. We agree that the word "only" at the end over-emphasizes the subject in the sentence. It is already clear from the beginning of the sentence that the data is being restricted to liquid rain subject in the sentence.
4. For clarity, we have added the labels of the panels of each Figure using lowercase alphabet.
5. The sentence in lines 233-234 had a typo, and we corrected it and rephrased it.
6. The small "k" for KDP is typo in line 256 and we corrected it.
7. The reference to the panels using lowercase alphabetic characters remains. We have added the labels on the upper right corner of each Figure (see question 4).

8. The information of the paragraph is indeed unnecessary. However, it has been shortened and rephrased so that the paragraph emphasizes that the analysis uses the parameter-optimized methods (those allowing it)

9. Thank you for notifying us of this typo. One of the words "using" has been removed.

10. The correct way is "its". We make the pertinent corrections

11. The correct way is "its". We make the pertinent corrections

12. We agree on removing "from methods" from the title of section 3.3

13. We appreciate the reviewer's suggestion regarding the inclusion of a statistical significance test. However, after careful consideration, we have concluded that such a test is not necessary nor appropriate in this context. The primary reason is that the analysis presented in this section does not involve testing a specific hypothesis or comparing distinct groups, which are typical prerequisites for conducting significance tests. Instead, our focus is on describing patterns and relationships between the KDP estimation methods, which are sufficiently addressed through the current method. As a result, we believe that adding this test would not enhance the value of the findings or contribute additional insights.

14. The word "plot" has been removed from the caption of Figure 15.

15. The sentence has been rephrased for better clarity.,

16. Indeed, we are referring to Figure 8.

17. The reference for Maesaka et a. (2012) is indeed incomplete. It has been updated.

**Changes in Manuscript:**

1. The suggested publication was included in the introduction, lines 17-18.

2. We changed "settings applied were" for "parameters used for the T-Matrix calculations were" in lines 109-110

3. We removed the word "only" in line 111.

4. Labels using lowercase alphabetic characters were used to indicate the panels in Figures 4, 6, 8, 10 and A1.

5. "Whereas" was removed and a new clearer sentence was added: "The underestimation of $K_{DP}$ using $10^5$ is evidenced in Fig. 5(b) for $Z_H \geq 35$ dBZ, where the results were the most negatively biased. The biases from the remaining parameters were equally consistent and smaller."

6. Change from "k" to "K" in line 256.

7. Reference labels to figures 4, 6, 8 and 10 were added.

8. The paragraph in lines 342-347 after "Performance Assessment of methods Relative to $K_{sc}^{DP}$" has been rephrased to emphasize that the analysis uses parameter-optimized methods (those allowing it): "The performance of the methods is analyzed qualitatively in Sec. 3.2.1 and quantitatively in Sec. 3.2.2. For these analyses, we used the parameter-optimized

kdp_maesaka, kdp_vulpiani, phase_proc_lp and kdp_from_phidp, and included kdp_schneebeli and kdp_iris."

9.  One of the words "using" has been removed in line 399

10. Changed "it's" for "its" in line 425

11. Changed "it's" for "its" in line 426

12. "from methods" was removed from the subsection title in line 433.

13. No changes.

14. The word "plot" has been removed from the caption of Figure 15

15. The sentence has been rephrased to: "It should be noted, however, that the reference framework and Giangrande et al. (2013)'s method use self-consistency relations to determine $K_{DP}$, and therefore, the uncertainties are correlated, and part of the reported performance is caused by this dependence"

16. The word "Figure" was added before "8"

17. The reference to Maesaka et al (2012) was updated as "conference paper" with the pertinent link to the conference paper.